# Host response to bacteria induces a shift towards the enteroendocrine cell lineage in the murine enteroid model

Marry Nissan[1], Adrienne Ranger[1], Justin J. Meade[1], Victoria Gillmore[2], Maija E. Lehn[1], Liliane Cabral-Fernandes[1], Derek K. L. Tsang[3], Scott D. Gray-Owen[2], Dana J. Philpott[1,3], Stephen E. Girardin[1,3]*

1 Department of Laboratory Medicine and Pathobiology, University of Toronto, Toronto, Ontario, Canada, 2 Department of Molecular Genetics, University of Toronto, Toronto, Ontario, Canada, 3 Department of Immunology, University of Toronto, Toronto, Ontario, Canada

* Stephen.girardin@utoronto.ca

## Abstract

The host response to commensal and pathogenic bacteria has been extensively characterized using human cancer cell line models but remains less defined in primary intestinal cellular systems. Recent evidence has demonstrated that mice lacking the Nod-like receptor (NLR) protein NLRC4 are susceptible to *Shigella flexneri* infection and thus represent a new model to study the mechanistic aspects of *S. flexneri*-host interaction. Using ileal organoids from wild-type (WT) and *Nlrc4⁻/⁻* mice, we first confirmed that NLRC4 was required for the restriction of intracellular *S. flexneri* growth. Surprisingly, NLRC4 further mediated the detection of bacteria-free *S. flexneri* supernatants, suggesting that ileal organoids sample proteins from the type three secretion system (T3SS) of *S. flexneri* to mediate a preemptive pyroptotic response to pathogens independently from invasion. Moreover, both invasive and non-invasive *S. flexneri* were found within *Nlrc4⁻/⁻* ileal organoids, suggesting that murine intestinal epithelial cells (IECs) may be capable of bacterial uptake. Transcriptional analysis further revealed that infection of *Nlrc4⁻/⁻* organoids with invasive or non-invasive *S. flexneri* resulted in the downregulation of inflammatory signaling. In addition, infection was associated with an enrichment for markers of the enteroendocrine cell (EEC) lineage, effects that required exposure to bacteria and were not recapitulated by bacteria-free supernatants. Together, our results reveal unexpected characteristics of host-bacterial interaction in primary murine IECs, which may shape the response to the microbiota and enteric pathogens at the intestinal mucosal surface.

**Data availability statement:** Bulk RNA sequencing data can be accessed from the GEO repository under the accession number GSE305881.

**Funding:** M.N. was supported by scholarships from the Canada Graduate Research Scholarship program and the Emerging and Pandemic Infections Consortium (EPIC), and this project was funded by grants from the Canadian Institutes for Health Research (CIHR) and Crohn's Colitis Canada (CCC) to S.E.G., and D.J.P The funders had no role in study design, data collection and analysis, decision to publish, or preparation of the manuscript https://epic.utoronto.ca/, https://crohnsandcolitis.ca/.

**Competing interests:** The authors have declared that no competing interests exist.

## Author summary

*Shigella* infections cause over 200,000 deaths each year, with the greatest impact in low-resource settings. The intestinal lining, made up of epithelial cells, forms the first barrier against invading bacteria—making it a critical target for studying host defense. In this study, we used murine intestinal organoids, lab-grown models of the gut epithelium, to examine how these cells respond to *Shigella flexneri* infection. By removing a key immune sensor (NLRC4), we could study bacterial entry and host responses without triggering immediate cell death. Surprisingly, even a supposedly harmless *Shigella flexneri* strain (BS176) entered the cells and suppressed inflammatory gene expression, despite lacking typical virulence factors. This suggests that intestinal epithelial cells may actively limit immune activation when no clear threat is detected. We also observed an increase in hormone-producing cells called enteroendocrine cells, pointing to a potential new and unrecognized role for these cells in sensing or responding to gut infections.

## Introduction

*Shigella flexneri* is a Gram-negative bacterium of the *Enterobacteriaceae* family that invades the human intestinal epithelium, causing inflammation and bacillary dysentery. Globally, *Shigella* is a major cause of diarrheal disease, particularly in children in low-resource settings, and its increasing resistance to antibiotics underscores the urgency of understanding its pathogenesis [1]. Due to its epithelial tropism, *S. flexneri* has become a widely used model pathogen for studying host-microbe interactions in the gut.

A key factor in *S. flexneri*'s virulence is its 220-kb virulence plasmid, which encodes a T3SS and an array of effector proteins that manipulate host cellular processes, resulting in host colonization upon traversing the intestinal epithelial layer [2]. Following adherence to the basolateral surface of epithelial cells, *S. flexneri* uses its T3SS—comprising a needle and rod complex—to inject effector proteins into the host cytosol, triggering membrane ruffling and bacterial uptake. Once inside, *S. flexneri* rapidly escapes the entry vacuole, gaining access to the cytoplasm, where it replicates, manipulates host signaling pathways and spreads from cell to cell.

Early studies investigating host detection of *S. flexneri* highlighted the initiation of pro-inflammatory signaling by infected cells [3–5]. One mechanism involves recognition of bacterial peptidoglycan (PGN) by the NLR proteins NOD1 and NOD2 in the host's cytosol leading to NF-κB activation and subsequent upregulation of cytokines and chemokines [6–12]. Other intracellular receptors, such as ALPK1, complement NOD1 and NOD2 by recognizing the bacterial metabolite ADP-heptose, a heptose phosphate intermediate in lipopolysaccharide (LPS) biosynthesis, and similarly trigger a robust NF-κB response [13–16]. Since cytokines are critical to the recruitment of immune cells such as monocytes, dendritic cells, and T cells to the site of infection,

it is unsurprising that *S. flexneri* has developed mechanisms to suppress this system. For example, the virulence effector protein IpaH$_{9.8}$ is an E3 ubiquitin ligase that targets NEMO, a key scaffold in the NF-κB pathway, for proteasomal degradation [17]. Additional effectors, including OspG and OspI, inhibit upstream E2 ligases necessary for IKK activation and IκBα degradation [18,19]. *S. flexneri* also dampens MAPK signaling via OspF, which irreversibly removes phosphate groups from ERK and p38 MAPKs, preventing downstream pro-inflammatory responses [20,21].

While these studies have provided valuable insights, most have relied on immortalized cancer-derived epithelial cell lines, which lack the spatial organization and genetic fidelity of the normal intestinal epithelium. To address these limitations, primary intestinal organoids—particularly enteroids derived from mouse or human intestinal crypts—have emerged as a valuable model for studying host-pathogen interactions. These 3D cultures recapitulate the crypt-villus architecture and harbour a heterogeneous population of epithelial cells, including LGR5 + stem cells, enterocytes, goblet cells, Paneth cells, tuft cells, and EECs [22,23]. This cellular system enables exploration of how bacterial pathogens, including *S. flexneri*, may affect epithelial differentiation and innate immune signaling in a more physiologically-relevantmodel.

Until recently, murine models were limited in their ability to support *S. flexneri* colonization due to robust epithelial immune defenses. Pioneering studies by Mitchell et al [24] and Roncaioli et al [25] identified the NAIP-NLRC4 inflammasome as a key innate defense mechanism in murine IECs, mediating pyroptotic cell death in response to the *S. flexneri* T3SS. This is evident by the fact that *Nlrc4*$^{-/-}$ mice, which lack this inflammasome, display increased susceptibility to infection, with elevated *S. flexneri* burden in IECs, cecal shortening, weight loss, and heightened chemokine expression when compared to their WT counterparts. These findings have established *Nlrc4*$^{-/-}$ mice as a tractable model for studying *S. flexneri* interactions with the intestinal epithelium. Building on this model, our study aims to interrogate how *S. flexneri* engages innate immune pathways and influences cell commitment within a physiologically-relevant epithelial context—an area that remains incompletely understood.

## Results

### Murine intestinal organoids elicit a preemptive response to virulent *S. flexneri* through NLRC4

To establish a murine enteroid infection model, we first evaluated pyroptotic cell death: a primary defense mechanism restricting *S. flexneri* infection in the murine intestinal epithelium. We infected Cultrex-dissociated WT murine ileal organoids with virulent (M90T) and plasmid-cured non-virulent (BS176) *S. flexneri* strains and assessed Gasdermin D (GSDMD) cleavage: a pore-forming protein that mediates pyroptotic cell death [26]. Only infection with M90T, but not BS176, induced GSDMD cleavage (Fig 1A), likely through the recognition of its T3SS needle and rod components by the NLRC4 inflammasome [27,28]. Interestingly, M90T and BS176 infection reduced full-length (p43) Caspase-11 protein levels (Fig 1A), suggesting that murine intestinal organoids also respond to *S. flexneri* independent of its virulence plasmid.

To further support the notion that GSDMD cleavage induced by M90T was linked to the detection of the T3SS, we used NeedleTox: a previously described chimeric protein made from the fusion of a T3SS needle protein and *B. anthracis* lethal factor (LFn), co-administered with anthrax protective antigen (PA) [27]. We found that NeedleTox induced GSDMD cleavage in an NLRC4-dependent manner (Fig 1B).

To establish a genetically tractable model for *S. flexneri* infection in murine enteroids, we next compared GSDMD between WT and *Nlrc4*$^{-/-}$ enteroids. Consistent with NLRC4-mediated recognition of T3SS proteins, deletion of *Nlrc4* abolished M90T-induced GSDMD cleavage (Fig 1C). Unexpectedly, exposure of enteroids to bacteria-free supernatants from M90T induced NLRC4-dependent GSDMD cleavage (Fig 1C), suggesting there is an internalization of bacteria-derived T3SS proteins for NAIP-NLRC4 activation. Activation of NLRC4 in this system was critical for the restriction of *S. flexneri* and the prevention of IEC colonization, as determined by colony-forming units (CFUs) quantification following gentamicin protection assay in WT and *Nlrc4*$^{-/-}$ enteroids (Fig 1D). Consistent with previous studies demonstrating a central role for NLRC4 in epithelial defence against *S. flexneri* [24,25], *Nlrc4*$^{-/-}$ enteroids had significantly higher CFUs of M90T than WT enteroids, likely reflecting reduced IEC pyroptotic cell death and enhanced intracellular bacterial persistence (Fig 1D).

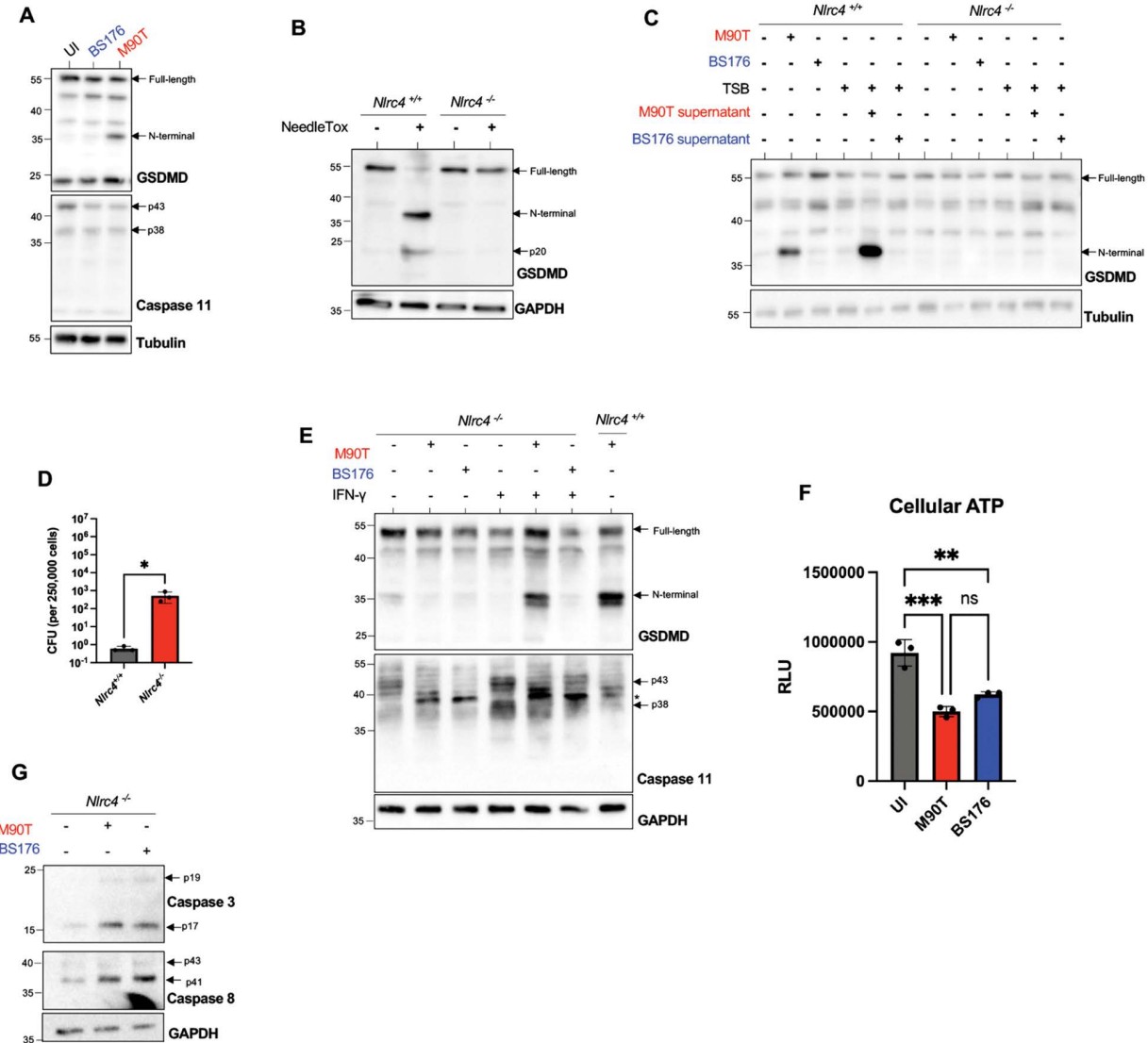

**Fig 1. Murine intestinal organoids elicit a preemptive response to virulent *S. flexneri* through NLRC4.** Murine ileal organoids were infected with virulent *S. flexneri* M90T or avirulent BS176 to assess NLRC4-dependent responses. (**A**) Western blot analysis of WT organoids infected with M90T or BS176. (**B**) Western blot of *Nlrc4+/+* and *Nlrc4-/-* organoids treated with 1 nM anthrax PA and 1 nM NeedleTox for 3.5 hours. (**C**) Western blot of *Nlrc4+/+* and *Nlrc4-/-* organoids infected with M90T or BS176, or treated with filtered bacterial supernatants. (**D**) CFUs recovered from *Nlrc4+/+* and *Nlrc4-/-* organoids following infection with M90T. (**E**) Western blot of *Nlrc4+/+* and *Nlrc4-/-* organoids primed with IFNγ (20 ng/mL, 18 h) before infection with M90T. Asterix denotes a non-specific band that appears with *S. flexneri* lysate. (**F**) Cellular ATP levels measured in *Nlrc4-/-* organoids infected with M90T or BS176. (**G**) Western blot analysis of *Nlrc4-/-* organoids infected with M90T or BS176 assessing apoptosis markers. Statistical significance in **D** was assessed by Student's t-test, and **F** by One-way ANOVA. Data represent mean±SD, with $p < 0.05$ considered statistically significant. All infections were performed at an MOI of 50 for 4 hours. Data are representative of at least three independent biological replicates.

In the absence of NLRC4, GSDMD cleavage can be driven by Caspase-11 upon detection of cytosolic LPS [29]. IFNγ is known to increase guanylate-binding protein (GBP) expression, which facilitates LPS binding to Caspase-11 in the cytosol [29,30]. Consistent with this, IFNγ primed *Nlrc4-/-* enteroids exhibited GSDMD cleavage in response to infection with the virulent M90T *S. flexneri*, but not with the avirulent strain BS176 strain (Fig 1E). These data indicate that, following IFNγ priming, virulent *S. flexneri* can engage Caspase-11 likely through T3SS-dependent access to the host cytosol.

To assess whether additional forms of cell death occur in the absence of IFNγ priming, we measured cellular ATP levels as a general readout of viability. Infection with both M90T and BS176 significantly decreased ATP in *Nlrc4*$^{-/-}$ enteroids (Fig 1F), indicating that infection triggers global cellular stress or death independently of inflammasome activation. We further evaluated apoptotic signalling and observed increased levels of cleaved Caspase-8 and cleaved Caspase-3 following infection with both strains (Fig 1G), suggesting that apoptosis contributes to IEC death in this model.

Collectively, our results support a model in which the murine IECs mount a preemptive, NLRC4-driven defense against *S. flexneri* through the detection of its T3SS components that occurs even in the absence of bacterial invasion. In contrast, bacteria lacking a T3SS fail to trigger NLRC4-dependent pyroptosis. Together with our subsequent findings that infection can induce apoptosis, these data suggest that IECs deploy multiple layers of defence that limit *S. flexneri* colonization. Despite this, *Nlrc4*$^{-/-}$ enteroids display a marked increase of intracellular bacterial persistence compared to WT controls, making them a suitable model to study host-pathogen interactions.

## Murine intestinal organoids exhibit non-selective bacterial uptake

We next asked whether, in the absence of NLRC4, M90T would display a marked advantage over BS176 in infecting enteroids, as expected from its virulence plasmid expressing host colonization factors. To address this, we compared bacterial burdens in *Nlrc4*$^{-/-}$ enteroids infected with M90T or BS176 to those observed in a standard cellular infection model: the human HeLa cell line. As expected, M90T achieved higher CFUs than BS176 at every timepoint in HeLa cells (Fig 2A). In striking contrast, CFU levels were similar between M90T and BS176 in *Nlrc4*$^{-/-}$ enteroids (Fig 2B and 2C), revealing a major difference in bacterial uptake and replication between this organoid system and classic infection models. Importantly, the relatively high CFU levels observed following BS176 infection in *Nlrc4*$^{-/-}$ enteroids suggest that even non-invasive bacteria are internalized by IECs as is consistent with a host-driven mechanism of bacterial uptake.

We next wanted to assess if this uptake phenomenon was specific to *S. flexneri*. *Nlrc4*$^{-/-}$ enteroids were treated with DH5α *E. coli*, and bacterial internalization was confirmed by CFUs following a gentamicin protection assay (Fig 2D). Transformation of *E. coli* with an afimbrial adhesin protein AfaE did not alter CFU numbers, suggesting bacterial uptake was not dependent on CD55 adhesion to the epithelium (Fig 2D). These findings were confirmed by visualization of intracellular *E. coli* by immunofluorescence (Fig 2E). Interestingly, we observed that the *Nlrc4*$^{-/-}$ enteroids could internalize 1-µm fluorescent beads, suggesting that particle uptake extended beyond live bacteria, and that this system was broadly permissive (Fig 2F).

To assess whether this phenomenon extended to other pathogen-associated molecular patterns (PAMPs), we treated WT enteroids with LPS. LPS failed to induce GSDMD p30 cleavage, even following IFNγ-mediated priming of Caspase-11 (Fig 2G). This lack of response is likely due to the hydrophobic nature of purified LPS and its inability to traverse the plasma membrane, suggesting that PAMP uptake in enteroids may exclude lipid-based cargo.

We finally sought to elucidate the mechanism by which non-invasive bacteria enter enteroids. Given their established role in bacterial translocation during antibiotic treatment, we investigated the contribution of Goblet cell-associated Antigen Passages (GAPs) [31–33]. During infection with either M90T or BS176, bacteria were observed to colocalize with goblet cells labelled by UEA1 in *Nlrc4*$^{-/-}$ enteroids (S1A Fig). Similarly, treatment with fluorescent beads also exhibited low-level colocalization with UEA1-positive cells (S1B Fig). To determine whether secretory cell expansion enhanced bacterial uptake, we treated enteroids with the γ-secretase inhibitor DAPT, which increases the IEC secretory cell pool. However, we observed no increase in bacterial translocation with DAPT, as CFU counts following infection remained comparable to untreated controls (S1C Fig). Interestingly, when using ISX-9, a small molecule used to skew enteroids to a secretory lineage that excludes goblet cells, there was an increase in CFUs following infection with M90T, but not with BS176 in *Nlrc4*$^{-/-}$ enteroids (S1C Fig). Consistent with their known effects on differentiation, DAPT treatment increased expression of *Muc2* and EEC markers, while ISX-9 preferentially increased expression of EEC markers along with tuft cell marker *Pou2f3* and Paneth cell marker *Lyz2* (S1D Fig) [34]. Collectively, these findings indicate that although bacteria and beads

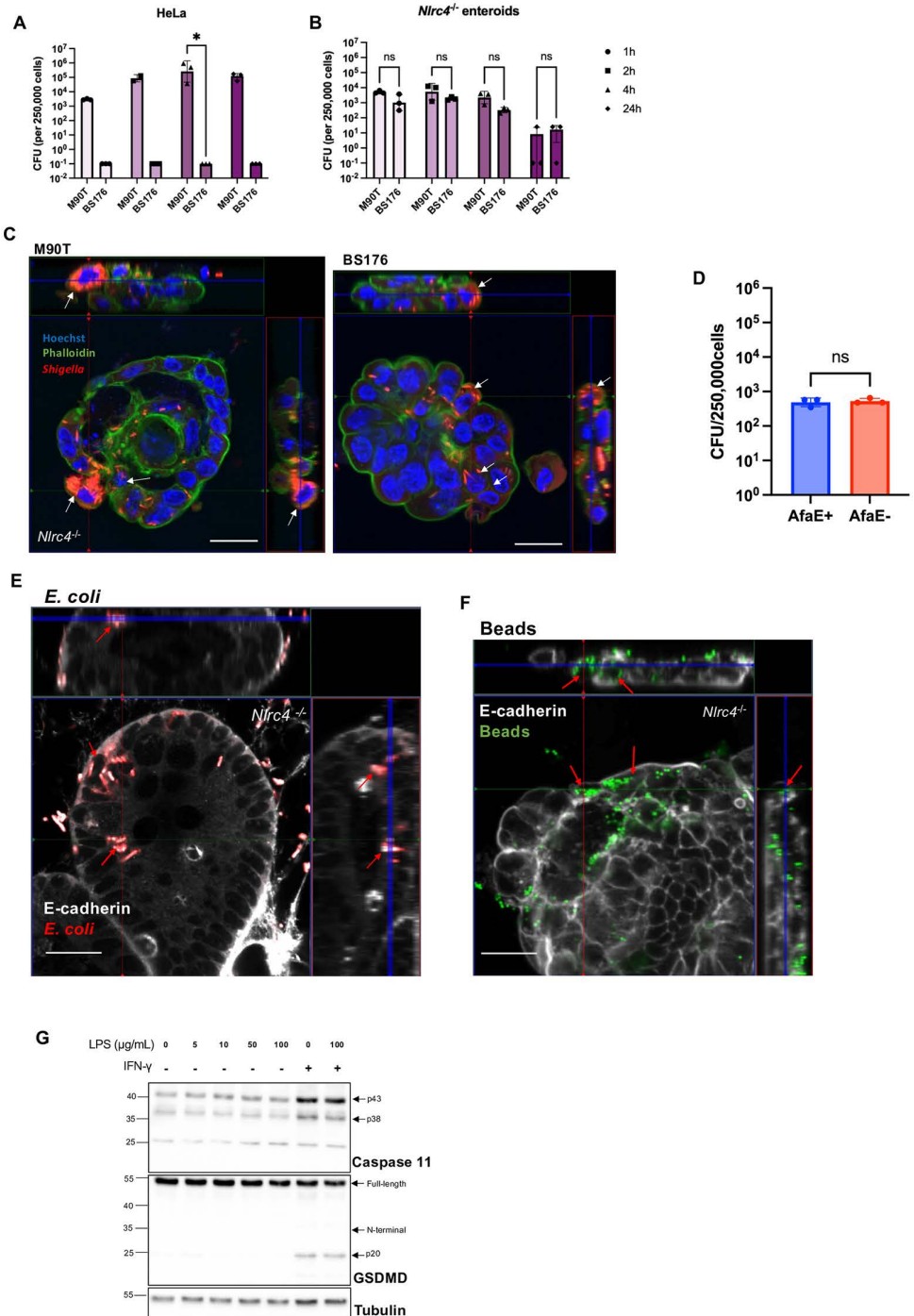

**Fig 2. Murine intestinal organoids exhibit non-selective bacterial uptake.** (**A**) CFUs from WT HeLa cells infected with M90T or BS176. (**B**) CFUs from *Nlrc4⁻/⁻* organoids infected with M90T or BS176. (**C**) Confocal immunofluorescence imaging of *Nlrc4⁻/⁻* organoids infected with M90T or BS176. (**D**) CFUs from *Nlrc4⁻/⁻* organoids infected with DH5α *E. coli.* transformed with an afimbrial adhesin protein AfaE. (**E-F**) Confocal immunofluorescence imaging of *Nlrc4⁻/⁻* organoids infected with DH5α *E.coli* (**E**) or fluorescent beads (**F**). (**G**) WT ileal organoids were stimulated for 4 hours with LPS. Where indicated, organoids were pretreated with IFNγ (20ng/mL) for 18 hours prior to stimulation. Protein expression was assessed by western blot. Confocal Images in **C, E** and **F** show an x–y projection (center), with orthogonal z–x and z–y views on the sides, compiled from a z-stack. Arrows point to cells that contain bacteria (**C, E**) or beads (**F**). Scale bar is 20 μM. All infections were performed at an MOI of 50 for 4 hours. Statistical analysis in panels **A** and **B** was performed using two-way ANOVA, and **D** by Student's t-test. Data represent mean ± SD, with $p < 0.05$ considered statistically significant. Data are representative of at least three independent biological replicates.

may associate with goblet cells, the secretory lineages do not appear to significantly mediate bacterial entry into enteroids in our model, and other modes of entry for non-pathogenic bacteria should be examined in future studies.

Taken together, these data demonstrate that murine IECs internalize material in a broadly permissive manner, including non-invasive *S. flexneri, E. coli*, and beads. Despite finding that M90T but not BS176 was likely capable of reaching the host cytosol following IFNγ priming (Fig 1E), our data suggests that cytosolic invasion remains relatively inefficient in murine IECs compared with conventional epithelial cells lines, as CFU recovery shows minimal differences between M90T and BS176 (Fig 2B). While some M90T may escape into the cytosol (as shown by GSDMD cleavage in IFNγ-primed enteroids), it may be a relatively rare event, especially in the absence of IFNγ priming.

### RIPK2 does not alter the transcriptome of murine intestinal organoids after acute *S. flexneri* M90T infection

Following the establishment of our enteroid infection model, we sought to investigate whether NLRs, besides NLRC4, contribute to the host response to *S. flexneri*. NOD1 was selected as a candidate due to its basal expression in IECs (Fig 3A) [16] and its previously described role in detecting *S. flexneri* in cell lines [6,7,10–12]. Since NOD2 is another cytosolic receptor that detects bacterial PGN and may play overlapping functions with NOD1 in IECs, we also considered its potential involvement, despite lower levels of expression at baseline in murine organoids (Fig 3A). Analysis of a single-cell RNA sequencing dataset recently published by our laboratory [35] confirmed that *Nod1, Ripk2* and to a lower extent *Nod2*, are constitutively expressed in the intestinal epithelium, spanning multiple epithelial cell types (Fig 3B). To address their involvement during infection, we utilized *Nlrc4⁻/⁻* enteroids, since infection of murine enteroids in NLRC4-competent cells does not support bacterial colonization (Fig 1). We generated *Nlrc4⁻/⁻Ripk2⁺/⁺* and *Nlrc4⁻/⁻Ripk2⁻/⁻* littermate mice, the latter lacking RIPK2, the adaptor kinase required for signaling by both NOD1 and NOD2.

To assess the contribution of RIPK2 in the epithelial response to *S. flexneri*, we performed bulk RNA sequencing on enteroids from both genotypes (*Nlrc4⁻/⁻Ripk2⁺/⁺* vs *Nlrc4⁻/⁻Ripk2⁻/⁻*) infected with M90T or treated with M90T bacterial supernatant (Fig 3C). Principal component analysis (PCA) and hierarchical clustering of gene expression profiles revealed that the major source of transcriptional variation was the infection itself, rather than the genotype (Fig 3D and 3E). Importantly, no genes passed the false discovery rate (FDR) threshold for differential expression between *Nlrc4⁻/⁻Ripk2⁺/⁺* and *Nlrc4⁻/⁻Ripk2⁻/⁻* organoids in either infection or supernatant treatment conditions (S2A and S2B Fig). Even among genes showing uncorrected nominal p-value significance, none exhibited consistent or biologically meaningful differences between the two genotypes when assessed for confirmation by RT-qPCR (Figs 3F and S2C). These data indicate that during acute infection with *S. flexneri* M90T, NOD1 and NOD2 signaling (via RIPK2) does not contribute to the modulation of the IEC transcriptional response.

### Neither NOD1 nor NOD2 stimulation with PGN elicits a broad pro-inflammatory response in murine intestinal organoids

As described above, the *S. flexneri* M90T virulence plasmid encodes several effector proteins that actively suppress NF-κB signalling. We hypothesized that immune suppression by *S. flexneri* may underlie the undetectable transcriptional contribution of NOD1 and NOD2 via RIPK2 during acute *S. flexneri* infection in organoids. Thus, to test whether enteroids can respond to NOD1 or NOD2 ligands in the absence of bacterial virulence factors, we stimulated WT murine ileal organoids with C12-iE-DAP or L18-MDP, which are canonical ligands for NOD1 and NOD2, respectively. Upon stimulation, however, we found no upregulation of pro-inflammatory genes (Fig 4A and 4B). Treatment with TNFα, a known inducer of NF-κB, robustly increased pro-inflammatory gene expression, confirming that the enteroids are competent to mount a transcriptional inflammatory response (Fig 4A and 4B). Given that these ligands likely exist in the lumen of the intestine along with the microbiota, we hypothesized that a mechanism for ligand entry may be optimized for the apical surface of IECs. To test this, we mechanically disrupted enteroids to expose their apical surface and then stimulated them with NOD1 and NOD2 ligands (S3A Fig). Similarly, we treated apical-out enteroids with NOD1 and NOD2 ligands (S3B Fig). However, neither approach elicited a pro-inflammatory response.

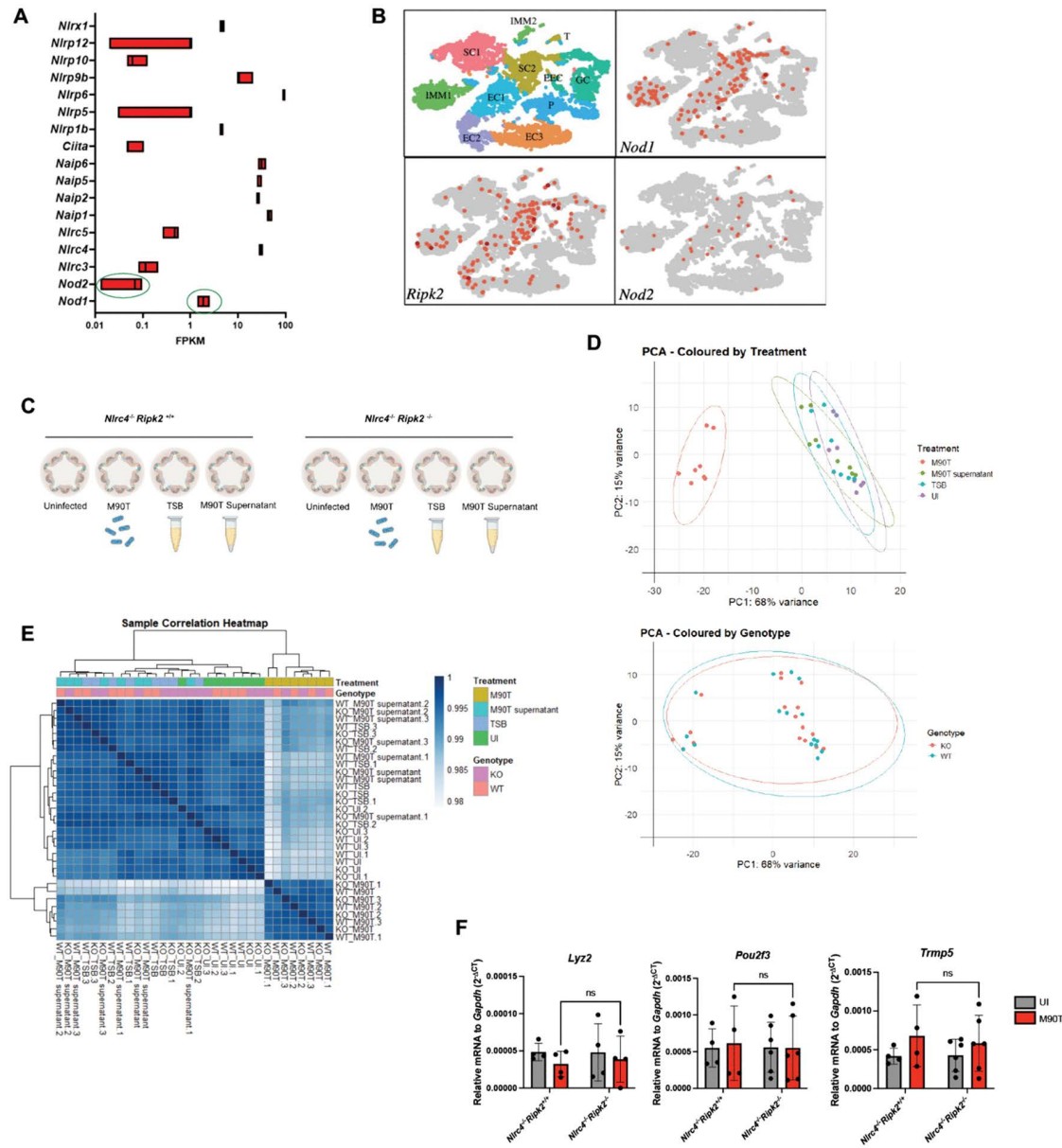

**Fig 3. RIPK2 does not alter the transcriptome of murine intestinal organoids after acute *S. flexneri* M90T infection.** (**A**) Relative expression (FPKM) of NLRs in WT murine ileal organoids under basal conditions, based on previously published bulk RNA sequencing data [GEO accession: GSE297523]. *Nod1* and *Nod2* are highlighted in green. (**B**) Uniform Manifold Approximation and Projection (UMAP) visualization of single-cell RNA sequencing data from WT murine ileal crypt-enriched epithelial cells, annotated by epithelial lineage identity [GEO accession GSE195742]. (**C**) Schematic representation of experimental design for bulk RNA sequencing of ileal organoids infected with *S. flexneri* M90T. (**D**) PCA of the top 500 differentially expressed genes, stratified by genotype and treatment condition. (**E**) Sample-to-sample correlation heatmap of the top 500 differentially expressed genes, showing clustering based on transcriptional relatedness. (**F**) qPCR validation in *Nlrc4⁻ᐟ⁻Ripk2⁺ᐟ⁺* and *Nlrc4⁻ᐟ⁻Ripk2⁻ᐟ⁻* ileal organoids of selected genes from that were statistically significant by *p*-value but did not pass FDR correction. Statistical significance in **F** was assessed using two-way ANOVA. Data represent mean ± SD, with $p < 0.05$ considered statistically significant. Panel **C** was created in BioRender. PHILPOTT, D. (2026) https://BioRender.com/yz630ln. All infections were performed using *S. flexneri* M90T at an MOI of 50 for 4 hours. Data are representative of at least four independent biological replicates.

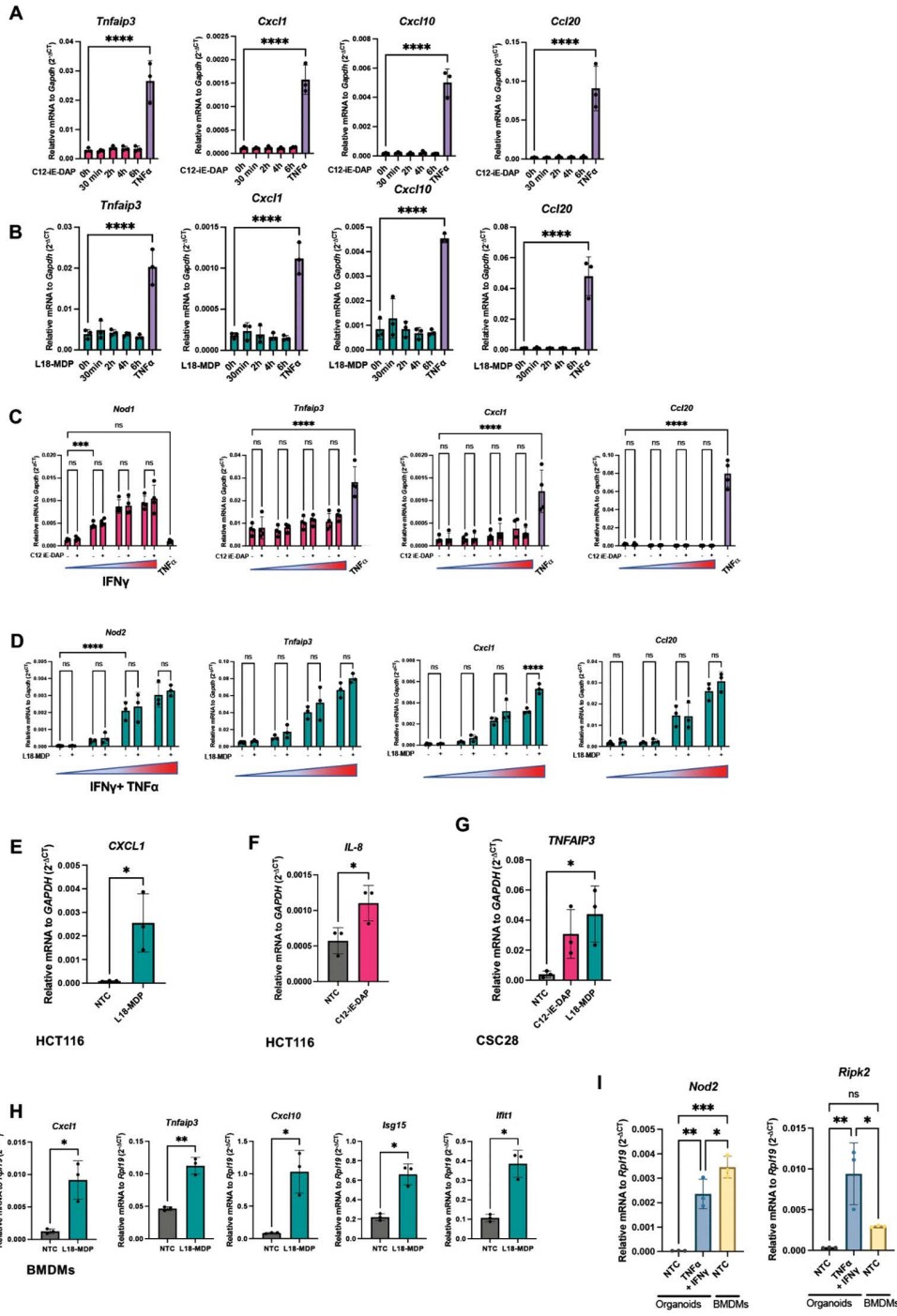

**Fig 4. Neither NOD1 nor NOD2 stimulation with PGN elicits a broad pro-inflammatory response in murine intestinal organoids. (A, B)** Time-course qPCR of pro-inflammatory gene expression in WT murine ileal organoids treated with 1 µg/mL C12-iE-DAP (**A**) or L18-MDP (**B**). TNFα (10 ng/mL, 4 hours) was included as a positive control. (**C**) qPCR of WT ileal organoids co-treated with C12-iE-DAP (1 µg/mL) and a titration of IFNγ (0.1, 1, 10,100ng/mL) for 4 hours. TNFα (10 ng/mL, 4 hours) was included as a positive control. (**D**) qPCR of WT ileal organoids co-treated with L18-MDP (1 µg/mL) and a titration of IFNγ+TNFα (0.1, 1, 10,100ng/mL) for 4 hours. (**E, F**) qPCR analysis of HCT116 cells treated with 1 µg/mL L18-MDP (**E**) or 1 µg/mL C12-iE-DAP (**F**) for 4 hours. (**G**) qPCR of CSC28 human colorectal cancer-derived cells treated with 1 µg/mL L18-MDP or 1 µg/mL C12-iE-DAP for 4 hours. (**H**) qPCR of BMDMs from WT mice stimulated with L18-MDP (1 µg/mL) for 4 hours. (**I**) Relative expression by qPCR of *Nod2* and *Ripk2* in matched WT BMDMs and ileal organoids derived from the same mouse. Statistical analyses were performed using one-way ANOVA for panels **A, B,**

G, and I; two-way ANOVA for panels **C** and **D**; and Student's *t*-test for panels **E**, **F**, **H**. Data represent mean ± SD, with $p < 0.05$ considered statistically significant. Data are representative of at least three independent biological replicates.

We next hypothesized that baseline NOD1 and NOD2 expression may be too low in enteroids to support downstream signaling. IFNγ stimulation sharply upregulated *Nod1* expression, but this increase did not enhance responsiveness to the NOD1 ligand, C12-iE-DAP, even at varying cytokine doses (Fig 4C). Similarly, co-stimulation with IFNγ and TNFα increased *Nod2* expression, yet exposure to the NOD2 ligand, L18-MDP, still failed to elicit a broad detectable inflammatory response (Fig 4D).

NOD1 and NOD2 responsiveness has previously been demonstrated in immortalized human epithelial lines, including HeLa and HCT116 cells [7–11]. In agreement with these reports, we observed a robust pro-inflammatory response to both C12-iE-DAP and L18-MDP in HCT116 and a patient-derived colorectal cancer line, CSC28 (Fig 4E–4G). This contrasts with the above findings, where primary murine enteroids failed to respond to the ligands, raising the possibility that NOD1 and NOD2 signaling is limited in this primary epithelial context. Interestingly, when we isolated bone marrow-derived macrophages (BMDMs) from WT mice and stimulated them with L18-MDP, BMDMs responded robustly – potentially due to either higher expression of *Nod2* or increased capacity to deliver the ligands into the cytosol in these cells compared to enteroids (Fig 4H and 4I) [36,37]. This revealed that primary murine cells are not intrinsically refractory to NOD1 and NOD2 stimulation, but rather that responsiveness is highly tissue- and cell-type-dependent.

Altogether, our findings suggest that NOD1 and NOD2 signaling pathways are functionally inactive in murine intestinal organoids under baseline conditions, although the underlying mechanism for this hyporesponsiveness as compared to human epithelial cancer cell lines or murine macrophages remains unclear.

## Downregulation of pro-inflammatory transcripts in response to *S. flexneri* infection occurs irrespective of virulence

Analysis of differential gene expression analysis between *S. flexneri* M90T-infected and uninfected *Nlrc4*⁻ᐟ⁻ enteroids revealed extensive transcriptional modulation. Notably, ~4,000 genes were downregulated upon infection, including numerous genes involved in canonical inflammatory signaling (Fig 5A). Among these were *Pmepa1*, *Dusp4*, *Plaur*, *Rela*, *Nfkb1*, *Cxcl16*, and *Ccl5* (Fig 5A). Intriguingly, this effect appeared to be specific to infection, as comparison between M90T and M90T supernatant-treated enteroids also showed downregulation of these same genes in the M90T-infected enteroids (Fig 5B). This suggests that the observed repression was likely not due to PAMPs shed into the media but rather required direct host-bacteria interaction. Unbiased pathway enrichment analysis further supported this observation, as "Hallmark TNFα signaling via NF-κB" was significantly downregulated in M90T-infected enteroids compared to uninfected controls (Fig 5C and 5D). This finding highlights an unexpected attenuation of NF-κB signalling below the levels observed in control enteroids during *S. flexneri* infection.

Given that M90T harbours a virulence plasmid encoding T3SS effectors known to suppress host inflammatory responses, we next asked whether this downregulation was virulence dependent. To address this, we compared gene expression in *Nlrc4*⁻ᐟ⁻ enteroids infected with either *S. flexneri* M90T or BS176. Surprisingly, both strains elicited a similar downregulation of key NF-κB targets (Fig 5E and 5F), indicating that this suppression was not solely attributable to virulence factor activity. Taken together, these results point to a broader, perhaps epithelial-intrinsic, mechanism of dampened immune signalling in IECs upon *S. flexneri* exposure that is independent of canonical virulence mechanisms.

## Terminally differentiated EECs are enriched following *S. flexneri* infection

To explore potential cell-type-specific responses to *S. flexneri*, we performed Gene Set Enrichment Analysis (GSEA) using defined gene sets for different cell types within the intestinal epithelium [38] on transcriptomic data comparing

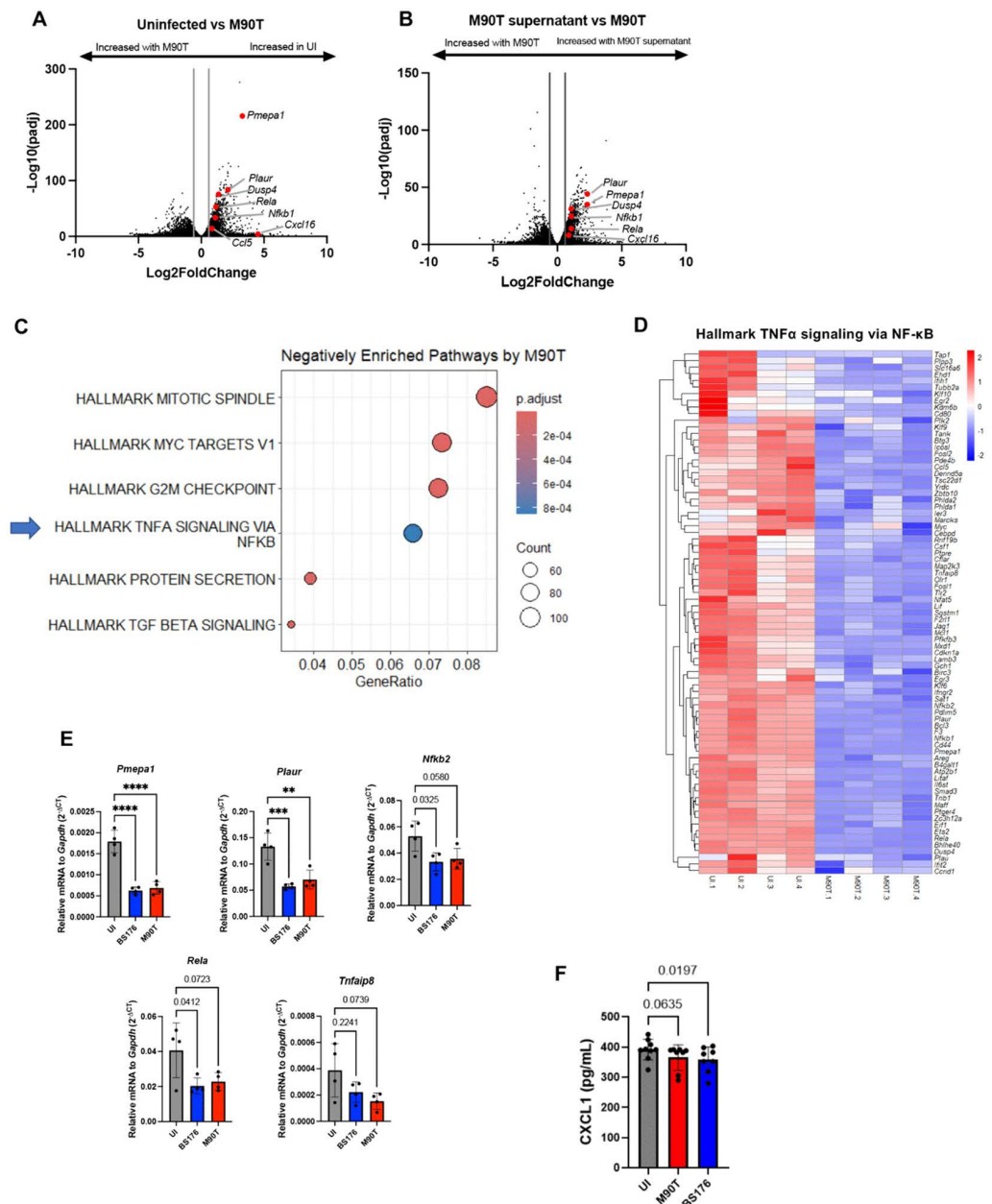

**Fig 5. *S. flexneri* suppresses inflammatory gene expression in IECs irrespective of virulence.** (**A**) Volcano plots showing differentially expressed genes between uninfected (UI) and *S. flexneri* M90T-infected *Nlrc4−/−* ileal organoids. Key pro-inflammatory genes are highlighted in red. (**B**) Volcano plots showing differentially expressed genes between *S. flexneri* M90T-infected and M90T supernatant-treated *Nlrc4−/−* ileal organoids. Key pro-inflammatory genes are highlighted in red. (**C**) Pathway enrichment analysis of the top downregulated pathways in *Nlrc4−/−* organoids infected with M90T versus UI controls, highlighting a suppression of inflammatory signaling terms. (**D**) Heatmap of genes within the "Hallmark TNFα signaling via NF-κB" gene set, based on variance-stabilizing transformed counts from UI and M90T-infected *Nlrc4−/−* organoids. (**E**) qPCR validation of selected inflammatory genes in *Nlrc4−/−* organoids infected with either M90T or BS176. (**F**) CXCL1 levels measured by ELISA in supernatants from *Nlrc4−/−* organoids infected with M90T or BS176. All infections were performed at an MOI of 50 for 4 hours. Statistical analysis in panels **E-F** were performed using one-way ANOVA. Data represent mean ± SD, with $p < 0.05$ considered statistically significant. Data are representative of at least four independent biological replicates.

uninfected and M90T-infected *Nlrc4-/-* enteroids. The most significant shift was the expansion of EECs after infection with M90T (Fig 6A). Interestingly, we also observed an upregulation of enterocyte and Paneth cell gene signatures after infection with M90T, while tuft and goblet cell signatures were downregulated (Fig 6A). Unbiased pathway enrichment analysis identified "Hallmark Pancreas Beta Cells" as one of the most significantly upregulated terms in infected *Nlrc4-/-* enteroids relative to uninfected controls (Fig 6B). Gene-concept network analysis revealed that many of the genes driving this term are canonical markers of EECs (Fig 6C). Indeed, several well-established EEC markers—including *ChgA*, *Neurod1*, and *Neurog3*—were significantly upregulated in response to infection, along with less characterized EEC-associated genes such as *Kcnt*, *Rimbp2*, *Miat*, and *Myt1* (Fig 6D).

To better characterize this shift, we utilized EEC-specific gene signatures to determine which cell types were affected with infection. Using subtype-specific gene signatures from Gehart et al [39], GSEA revealed a broad upregulation across most EEC populations, with the notable exception of L-cells, which remained unchanged (S4A Fig). To determine whether this shift reflects a bias toward early EEC commitment or terminal differentiation, we analyzed stage-specific EEC markers [39]. Infection led to increased expression of terminally differentiated EEC markers, while early EEC progenitor markers were downregulated (Fig 6E), suggesting that *S. flexneri* promotes terminal EEC differentiation rather than expansion of the EEC progenitor pool.

To determine whether this shift was virulence-dependent, we compared EEC marker expression in organoids infected with either M90T or BS176. Both strains increased EEC-associated transcripts, indicating that this effect does not require the virulence plasmid (Fig 6F). Indeed, even DH5α *E.coli* was able to upregulate these EEC-associated transcripts (S4B Fig). When assessing late EEC markers by immunofluorescence, both *S. flexneri* strains increased the number of ChgA-positive cells per organoid (Fig 6G), but only M90T increased the number of serotonin-positive cells per organoid (Fig 6H), suggesting that virulence may selectively modulate specific EEC subtypes.

Further experiments revealed that sonicated M90T, but not heat-inactivated M90T, induced EEC markers to levels comparable to live bacteria (Fig 6I). Coupled with the fact that EECs are not upregulated with bacterial supernatants (S4C Fig), these data suggest that intact, non-secreted molecules (possibly proteins) from *S. flexneri* are sufficient to modulate the EEC compartment. These results suggest that the murine intestinal epithelium skews lineage commitment toward mature EEC subsets following bacterial detection, potentially as a strategy to modulate host sensory or secretory responses at the mucosal interface. Altogether, these data lay the groundwork for future studies aimed at identifying the bacterial factors responsible for driving EEC differentiation and elucidating their functional impact on host physiology. A schematic summary of the proposed model is provided in S5 Fig.

## Discussion

Enteric infections remain a major cause of morbidity and mortality worldwide, particularly among young children and individuals in low-resource settings. *Shigella* is estimated to cause over 200,000 deaths annually, highlighting the urgent need to better understand intestinal epithelial immune responses to infection [40]. In addition, gut pathologies such as inflammatory bowel disease (IBD) are often associated with microbiota dysbiosis, particularly an overgrowth of *Enterobacteriaceae*. Indeed, increased abundances of *Shigella* spp. and *Escherichia coli* have been linked to IBD and IBD treatment failure [41]. Alongside these expansions of *Enterobacteriaceae,* patients typically exhibit increased secretion of pro-inflammatory cytokines (e.g., TNFα, IL-6), loss of barrier function, reduced production of short-chain fatty acids, and diminished bacterial diversity [41]. Given that IECs form the first line of defense against pathobionts, understanding their role in host defense is critical.

Using *Nlrc4-/-* enteroids that are permissive to infection due to a lack of pyroptosis, we sought to investigate the global transcriptional response of murine intestinal organoids to *S. flexneri* infection. During the establishment of our model, we found that virulent *S. flexneri* M90T induced GSDMD cleavage independently of bacterial invasion and that the introduction of sterile-filtered bacterial supernatant from the virulent strain was sufficient to elicit a pyroptotic

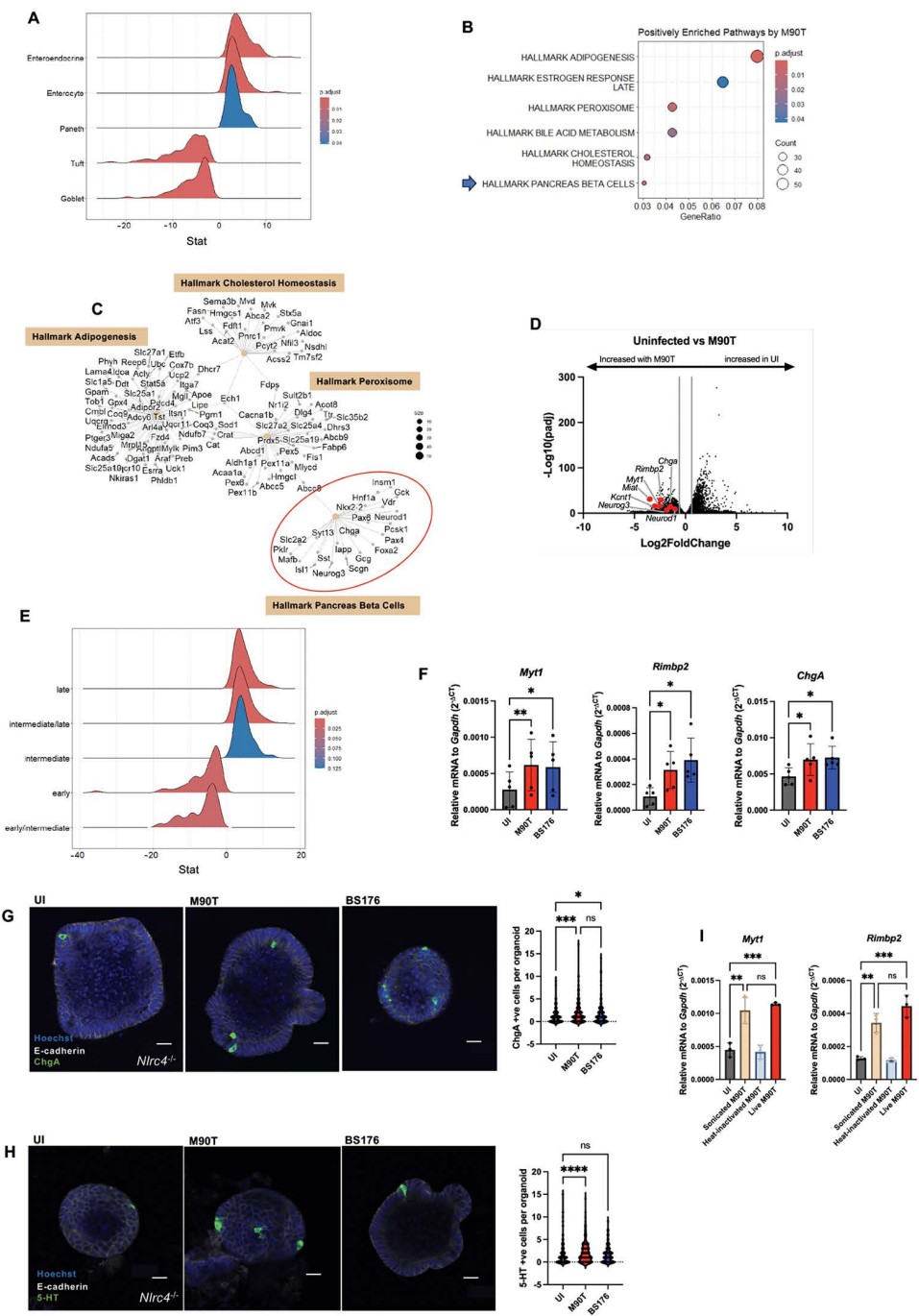

**Fig 6. Terminally differentiated EECs are enriched in response to *S. flexneri* infection.** (**A**) GSEA of IEC type signatures defined by Haber et al. [38], comparing *Nlrc4⁻ᐟ⁻* organoids infected with M90T versus UI controls. (**B**) Pathway enrichment analysis of the top upregulated gene sets in *Nlrc4⁻ᐟ⁻* organoids following M90T infection versus UI controls. Arrow points to the "Hallmark Pancreas Beta Cells" gene set. (**C**) Gene-concept network plot showing individual genes contributing to top enriched pathways from panel B, with the "Hallmark Pancreas Beta Cells" gene set circled. (**D**) Volcano plot of differentially expressed genes between UI and *S. flexneri* M90T-infected *Nlrc4⁻ᐟ⁻* ileal organoids, with key EEC markers highlighted in red. (**E**) GSEA showing differential enrichment of EECs at various stages of maturation defined by Gehart et al [39] between M90T-infected and UI *Nlrc4⁻ᐟ⁻* enteroids. (**F**) qPCR validation of selected EEC markers in *Nlrc4⁻ᐟ⁻* organoids infected with M90T or BS176. (**G, H**) Confocal immunofluorescence staining for ChgA (**G**) and 5-HT (**H**) in *Nlrc4⁻ᐟ⁻* organoids infected with M90T or BS176, with quantification of number of ChgA⁺ cells (**G**) and 5-HT⁺ cells (**H**) per organoid. Scale bar represents 20 um. (**I**) qPCR analysis of EEC markers in *Nlrc4⁻ᐟ⁻* organoids treated with live M90T, sonicated M90T, or heat-inactivated M90T. Bacterial

input was stoichiometrically matched across all conditions. All infections were performed at an MOI of 50 for 4 hours. Statistical analysis in panels **F**, **G**, **H** and **I** was performed using one-way ANOVA. Data represent mean ± SD, with $p < 0.05$ considered statistically significant. Data are representative of at least three independent biological replicates.

response in organoids (Fig 1A and 1C). This suggests that PAMPs were likely taken up by IECs for detection by the intracellular sensor NLRC4. The uptake of the external milieu by the organoids extended beyond PAMPs, as even non-virulent *S. flexneri* BS176, *E. coli*, and fluorescent beads were able to gain entry into the organoids (Fig 2). Taken together, these findings suggest that the murine intestinal epithelium may possess the ability to sample large cargo from its external environment.

One intriguing finding is that both M90T and BS176 were internalized by enteroids at comparable levels (Fig 2B), indicating that virulence factors do not provide a strong entry advantage in this system. This contrasts with HeLa cells, where M90T strikingly outcompetes BS176 (Fig 2A), suggesting that *S. flexneri* invasion mechanisms may be human-adapted and lack a selective advantage in murine models. Notably, IFNγ priming revealed GSDMD cleavage in *Nlrc4*<sup>-/-</sup> enteroids infected with M90T, but not BS176 (Fig 1E). IFNγ priming has been reported to facilitate Caspase-11 binding to intracellular LPS and increase both GBP and Caspase-11 expression [30]. Since the T3SS of M90T enables vacuolar escape and exposure of LPS to the Caspase-11 detection system, our data is consistent with the known physiology of the pathogen and suggests that Caspase-11 can partially compensate for the absence of NLRC4 in IFNγ-primed cells. These observations align with previous reports that IFNγ contributes to intestinal protection by restricting *S. flexneri* replication in IECs [42].

Despite these differences, CFU recovery remained similar between M90T and BS176 in *Nlrc4*<sup>-/-</sup> enteroids, indicating that cytosolic invasion does not translate into major growth advantage in this model. These results suggest that cytosolic invasion is relatively inefficient in murine IECs and that additional host or microbial factors may be required for intracellular replication.

A potential limitation of our study is that the murine small intestinal organoid model may not fully recapitulate human colonic infection, which is the primary site of shigellosis in humans. Indeed, some bacterial effectors may only target human proteins. However, this potential limitation also comes with benefits as our experimental setting allows for better characterizing the underpinnings of host-driven responses in the absence of pathogen-driven manipulation. Nevertheless, future work aiming to validate key findings in human colonic organoids to better reflect the natural tropism of *S. flexneri* will be useful to complement and further validate our findings.

Antigen sampling in the small intestine is typically attributed to M cells, which have been proposed as an entry point for *S. flexneri* [43–45]. However, since organoids lack M cells in the absence of RANKL stimulation [46], our model suggests the presence of an alternative entry mechanism. As a possible candidate, we explored the role of GAPs, which have been reported to mediate luminal sampling via fluid-phase endocytosis during mucus secretion as a tolerogenic-promoting response [31,32]. Notably, bacteria have been shown to exploit GAPs to breach the intestinal barrier [33]. In our model, however, increasing the number of goblet cells in *Nlrc4*<sup>-/-</sup> enteroids did not result in increased bacterial translocation (S1C Fig) Although treatment with ISX-9 increased M90T counts in *Nlrc4*<sup>-/-</sup> enteroids, there was no corresponding increase observed with BS176, suggesting other secretory cells do not serve as a primary route for bacterial uptake in this context either (S1C Fig). These findings raise the possibility that downstream host sensing and immune activation may contribute to the differential control of virulent *S. flexneri* strains.

Due to their expression in the intestinal epithelium and their reported role in restricting *S. flexneri* in cancer cell lines [6–12], we investigated the roles of NOD1 and NOD2 in ileal organoids. Surprisingly, transcriptomic analyses revealed no significant differences between *Nlrc4*<sup>-/-</sup>*Ripk2*<sup>+/+</sup> and *Nlrc4*<sup>-/-</sup>*Ripk2*<sup>-/-</sup> enteroids post-infection and we were also unable to trigger pro-inflammatory signaling in response to synthetic NOD1/NOD2 ligands (Figs 3 and 4). To address whether this

lack of responsiveness was due to low expression of PRRs, we primed organoids with IFNγ or IFNγ+TNFα, which effectively increased *Nod1* and *Nod2* expression (Fig 4C and 4D). This, however, failed to restore responsiveness, arguing against a purely transcriptional limitation. Given that IECs have evolved highly regulated sampling mechanisms to maintain barrier integrity and limit inappropriate immune activation, it is plausible that primary IECs either lack a mechanism to deliver NOD1/NOD2 ligands into the cytosol or actively restrict this process as a means of preserving immune tolerance. Such compartmentalization may represent an evolved strategy to limit cytosolic microbial sensing to contexts in which additional danger signals, such as epithelial damage, override the default tolerogenic state. Future studies will be necessary to identify the cellular or molecular mechanisms that distinguish NOD1/NOD2 responsiveness between immune cells, transformed epithelial cell lines, and primary IECs.

In examining the global transcriptional response to *S. flexneri*, we observed downregulation of NF-κB targets (Fig 5). Classically, *S. flexneri* downregulates inflammation via virulence effectors that target host signaling through mechanisms such as dephosphorylation, ubiquitination, and deubiquitylation. However, we found that even BS176, the plasmid-cured strain of *S. flexneri*, suppressed pro-inflammatory signaling similarly (Fig 5E and 5F), suggesting that this response is not strictly virulence-dependent. Sperandio et al. [12] reported that *S. flexneri*, even when non-invasive, can downregulate several immune genes (e.g., *CXCL11*, *IL1R2*, *IL18*, *IL20RA*, *HBD2/3*). Still, the prevailing view holds that *S. flexneri* elicits a strong pro-inflammatory response in epithelial cells. Our findings challenge this paradigm. The result that even non-invasive bacteria can enter *Nlrc4*$^{-/-}$ organoids and elicit transcriptional repression supports a model of host-mediated tolerance. In this model, the epithelium would downregulate inflammatory signaling during bacterial sampling to prevent inappropriate immune activation. Taken together, the widespread transcriptional downregulation we observe, even in response to non-virulent *S. flexneri*, may reflect a host-intrinsic strategy to sample the extracellular milieu and downregulate inappropriate inflammatory signaling in the absence of virulence detection. By deleting NLRC4, the primary sensor of *S. flexneri* T3SS, our model may have shifted toward a more sampling-forward, tolerogenic state—one in which epithelial cells default to immune quiescence when virulence cues are absent or undetectable.

Additional exploration of the transcriptome of infected organoids revealed an expansion of EECs, suggesting a shift in cell commitment upon microbial sensing (Fig 6). Interestingly, our data show that EEC commitment occurs independently of virulence, as it is observed with both M90T and BS176 (Fig 6F and 6G). Moreover, sonicated, but not heat-inactivated, M90T was sufficient to recapitulate the phenotype (Fig 6I). Importantly, our sequencing data revealed that EECs were not upregulated in response to M90T supernatants (S4C Fig), suggesting that structural or non-secreted protein(s) of *S. flexneri* may mediate this shift in cell commitment. GSEA further revealed that infection did not upregulate early markers of EEC differentiation. In fact, their expression was reduced, suggesting that infection influences the maturation of existing EECs rather than initiating new lineage commitment (Fig 6E).

EECs are hormone-secreting cells that play an essential role in microbial sensing and intestinal homeostasis. Hormones produced by EECs, such as 5-HT, have been shown to promote the activation of macrophages and maturation of lymphocytes [47,48]. EECs also differentially express *Piezo2*, a mechanosensitive ion channel, enabling vesicle release upon mechanical disruption [49–51]. In addition, EECs have been shown to promote tight junction protein expression (e.g., ZO-1, Occludin) following GLP-2 release [52]. Taken together, these features suggest that EECs are not only sensors of microbial stimuli but also key communicators with immune cells, positioning themselves as critical modulators of infection outcomes. Together, these findings suggest that terminally differentiated EECs may play a previously underappreciated role in host-microbe interactions. Whether this shift reflects enhanced hormonal output, modulation of microbial sensing, or feedback to local immune populations remains an open question.

## Methods

### Ethics statement

The Animal Ethics Review Committee of the University of Toronto approved all animal experiments

## Reagents for treatments

Reagents for treatments are as listed: L18-MDP (Invivogen, tlrl-lmdp), C12-iE-DAP (Invivogen, tlrl-c12dap), Tri-DAP (Invivogen, tlrl-tdap), TNF (ThermoFisher, 315-01A-5UG), IFNγ (ThermoFisher, PMC4031), Fluoresbrite YG Carboxylate Microspheres 1.00µm (Polysciences, 15702), DAPT (abcam, ab120633), ISX-9 (Tocris, 4439), Ultrapure LPS, *E. coli* 0111:B4 (InvivoGen, tlrl-3pelps), and NeedleTox and PA were a gift from Dr. Jeremy Mogridge (University of Toronto, ON).

## Mice

WT C57BL/6J mice were originally obtained from the Jackson Laboratories. *Nlrc4*^-/- mice were kindly obtained from the lab of Dr. Russell Vance (University of California, Berkeley) [53]. *Nlrc4*^-/- mice were genotyped by PCR using the primers: forward 5'-ATGGGTCCAGCATGAACGAG- 3' and reverse 5' TCTGAGAACAAATTGATGCCACAC-3'. Restriction enzyme digestion using BmpI (NEB, R0565) was carried out on the PCR products to identify the mutant nucleotide insertion. *Ripk2*^-/- mice were kindly obtained from Dr. Richard Flavell (Yale University School of Medicine) and were backcrossed to a C57BL/6 background [54]. *Ripk2*^-/- mice were genotyped by PCR using the primers: common forward 5'- TTG GAG CTT CCT CTA GTG CTG- 3', WT reverse 5'-TGC AAA GTG ATG TGA CTG AAT G-3', and mutant reverse 5'-CCT TCT ATC GCC TTC TTG ACG-3'. *Nlrc4*^-/-*Ripk2*^-/- mice were generated by crossing *Nlrc4*^-/- and *Ripk2*^+/- mice. All mice were maintained in a specific pathogen-free facility with a 12-hour light-dark cycle. Mice were given acidified water and irradiated chow. Organoids used for this paper were created with littermates and sex-matched mice.

## Murine organoid culture

Murine intestinal organoids were generated as previously described [22,55]. Briefly, the ileum was isolated from mice and gently scraped with a glass coverslip to remove excess mucus, feces, and villus tips. The tissue was then cut into small segments and washed three times with Dulbecco's phosphate-buffered saline (D-PBS; Wisent, 311–425-CL). Following the final wash, the ileal fragments were incubated in 2 mM EDTA (Bioshop, EDT111.100) in D-PBS for 30 minutes at 4°C to facilitate crypt release.

After incubation, crypts were dislodged by vigorously shaking the fragments in fresh D-PBS, and the supernatant containing the crypts was filtered through a 70 µm cell strainer (Fisherbrand, 22-363-548). The collected crypts were pelleted by centrifugation at 500×g for 5 minutes and washed three times with Advanced DMEM/F12 (Wisent, 2634028) supplemented with 1×Penicillin-Streptomycin (Wisent, 450–201-EL) to kill bacteria.

Washed crypts were resuspended in Cultrex Basement Membrane Extract, Type 2 (R&D Systems, 3532-010-02) and allowed to polymerize at 37°C for 15 minutes. The polymerized domes were overlaid with organoid growth medium composed of Advanced DMEM/F12 supplemented with 1×Penicillin-Streptomycin, 1×GlutaMAX (Gibco, 35050–061), 10 mM HEPES (ThermoFisher, 15630080), 10% (v/v) R-spondin conditioned media, 10% (v/v) Noggin conditioned media (both prepared in-house), 1×B27 Supplement (Gibco, 17504044), 1×N-2 Supplement (Gibco, 17502048), 1 mM N-acetylcysteine (Sigma, A9165-100G), 50 ng/mL recombinant mouse EGF (ThermoFisher, PMG8041), and 10 µM Y-27632 ROCK inhibitor (Tocris, 1254). Y-27632 was removed from the culture medium three days post-passage or after initial crypt embedding.

Organoids were maintained at 37°C in a humidified 5% $CO_2$ incubator and passaged every 7 days. Media was changed 3 days after seeding the crypts. For passaging, organoids were gently disrupted in washing medium (D-PBS supplemented with 10% heat-inactivated fetal bovine serum (FBS; Wisent, 098150) using a P10 pipette tip fitted onto a 5 mL serological pipette. The resulting fragments were washed three times with washing medium and centrifuged at 70×g for 3 minutes at each step.

## Cell lines

HeLa, HCT116, and CSC28 cells were used in this study. CSC28 cells were kindly obtained from Dr. Catherine O'Brien (University of Toronto, Toronto). HeLa and HCT116 cells were cultured in Dulbecco's Modified Eagle Medium (DMEM;

Wisent, 319–005-CL) supplemented with 10% heat-inactivated FBS and 1 × Penicillin-Streptomycin. CSC28 cells, a patient-derived colorectal cancer cell line, were maintained in suspension flasks (Sarstedt, 83.3911.502) using DMEM/F12 as a base medium. This was supplemented with 0.2% (v/v) lipid concentrate (Sigma, L0288), 4 µg/mL heparin (Sigma, H3149-100KU), 0.5% (v/v) Fungizone (Gibco, 15290018), 0.4% (v/v) NeuroCult SM1 Neuronal Supplement (STEMCELL Technologies, 05711), 1 × N-2 Supplement, 1 × Non-Essential Amino Acids (Hyclone, SH302380), 1 mM sodium pyruvate (Hyclone, SH3023901), 10 mM HEPES, 1 × GlutaMAX, 1 × Penicillin-Streptomycin, 20 ng/mL epidermal growth factor (PeproTech, AF-100–15), and 10 ng/mL fibroblast growth factor (STEMCELL Technologies, 78003.1).

HeLa and HCT116 cells were passaged at confluency using Trypsin-EDTA (Wisent, 325–043-EL) to detach the cells, followed by inactivation with complete DMEM containing 10% FBS. CSC28 cells were passaged every seven days. To do so, spheroids were first pelleted by centrifugation at 500 × g for 5 minutes, then incubated with Trypsin-EDTA at 37°C for 30 minutes until dissociated into single cells. Trypsin was inactivated by adding DMEM + 10% FBS, and cells were centrifuged again at 500 × g for 5 minutes. The resulting pellet was resuspended in fresh CSC28 suspension media. Cells were grown at 37°C in 5% $CO_2$.

### BMDM culture

BMDMs were prepared as previously described [56]. Briefly, femurs and tibias were collected from mice aged 8–10 weeks. Bone marrow was obtained by flushing ice-cold RPMI 1640 (Wisent, 350–000-CL) through the bones using a 24-gauge needle. Red blood cells were removed with ACK lysis buffer (Gibco, A10492-01), and the resulting cell suspension was passed through a 50 µm filter. Cells were plated in 10 cm² dishes containing complete RPMI 1640 supplemented with L929 conditioned medium (made in-house) and cultured until confluent. Prior to treatment, BMDMs were counted, seeded into 24-well plates at a density of $1 \times 10^6$ cells/well, and allowed to adhere overnight.

### Bacteria strains and cultivation

*Shigella flexneri* strains M90T and BS176 were used in this study. Bacteria were maintained on tryptic soy broth (TSB) agar plates supplemented with 1% Congo red and incubated overnight at 37°C. For M90T, red colonies were selectively picked to ensure retention of the virulence plasmid encoding the T3SS.

Liquid overnight cultures were initiated by inoculating a single colony into TSB and incubating at 37°C with shaking at 225 rpm. For day cultures, overnight cultures were diluted 1:100 in fresh TSB and grown at 37°C with shaking until mid-exponential phase, defined as an optical density at 600 nm ($OD_{600}$) of 0.4–0.6.

Before infection, bacteria were pelleted by centrifugation at room temperature (6000x g, 5 minutes), washed once with D-PBS, and resuspended in the appropriate infection medium. For organoid infections, bacteria were suspended in organoid infection medium (organoid growth medium lacking Y-27632 and Penicillin-Streptomycin). For HeLa cell infections, bacteria were suspended in HeLa infection medium (DMEM supplemented with 10% heat-inactivated FBS lacking Penicillin-Streptomycin).

DH5α *E.coli* (Invitrogen, 18265017) was used in this study. Bacteria were maintained on LB agar plates and incubated overnight at 37°C. Liquid overnight cultures were initiated by inoculating a single colony into LB and incubating at 37°C with shaking at 225 rpm. For day cultures, overnight cultures were diluted 1:100 in fresh LB and grown at 37°C with shaking until mid-exponential phase, defined as an optical density at 600 nm ($OD_{600}$) of 0.4–0.6. Preparation for infection was done similarly to *S. flexneri* strains mentioned above.

### Infection of organoids

Organoids were cultured in organoid growth media with 10 µM Y-27632 for 3 days post-passage and cultured for an additional day in organoid growth media without Y-27632. Before infection, day 4 organoids were removed from Cultrex using 300µL of

Cultrex Organoid Harvesting per dome (Biotechne, 3700-100-01) and gently rotated at 4°C for 20 minutes. Organoids were spun down (70x g, 5 minutes),washed with D-PBS and suspended in organoid infection media containing bacteria to achieve a multiplicity of infection (MOI) of 50. Cell counts for the MOI calculations were done by applying trypsin-EDTA to one organoid dome per genotype and counting cells using a TC20 automated cell counter (Bio-Rad). Upon bacterial application, organoids were spinfected at 1500 x g for 5 minutes and allowed to incubate with the bacteria for 30 minutes. Following this, organoids were washed twice with gentamicin-containing organoid infection media (Sigma, G1397-10ML) and resuspended in 30ul domes of Cultrex containing 50 µg/mL of gentamicin. Domes were allowed to polymerize for 10 minutes and overloaded with 500µL of gentamicin-containing infection media to kill extracellular bacteria. Gentamicin was used at 50µg/mL at each step. At 4 hours, organoids were lysed or fixed according to the assay protocol. All infections were done at 37°C in 5% $CO_2$. Littermate and sex-matched mice were used for organoid preparation when making comparisons between genotypes.

M90T supernatants were prepared by sterile filtering liquid cultures of *S. flexneri* using a 0.2 µm filter and applying that to organoids at a 1:1 ratio with organoid infection media. Filtered supernatants were plated on TSB agar plates to ensure no bacterial growth. Bacterial lysates were prepared by suspending the day culture of *S. flexneri* with organoid infection media and sonicating. Heat-inactivated bacteria were prepared by incubating the washed *S. flexneri* pellet in organoid infection media at 85°C for 30 minutes. The heat-inactivated M90T solution was plated on TSB agar plates to ensure no bacterial growth.

### Infection of HeLa cells

HeLa cells were passaged one day prior to infection using Trypsin-EDTA, followed by neutralization with DMEM supplemented with 10% heat-inactivated FBS. Cells were counted using a TC20 automated cell counter (BioRad), and 250,000 cells were seeded per well in 6-well tissue culture plates (ThermoFisher, 140675). The following day, cells were infected in HeLa infection medium (DMEM + 10% FBS) at an MOI of 50. To facilitate bacterial contact, plates were centrifuged at 1,500 × g for 5 minutes and incubated at 37°C for 30 minutes.

After incubation, cells were washed twice with gentamicin-containing infection medium (50 µg/mL) and then overlaid with 500µL of fresh gentamicin medium to kill extracellular bacteria. At 4 hours post-infection, cells were washed twice with D-PBS and lysed in 0.1% Triton X-100 in D-PBS for CFU determination.

### Apical-out organoid culture

Apical-out organoids were generated as previously described with modifications [57]. Briefly, organoids were cultured in organoid growth media with 10 µM Y-27632 for 3 days post-passage. Day 3 organoids were removed from Cultrex using 300µL of Cultrex Organoid Harvesting per dome and gently rotated at 4°C for 20 minutes. Organoids were gently spun down at 70 x g for 5 minutes at 4°C and resuspended in organoid growth media without Y-27632 for 2 days in ultra-low attachment plates (Corning, 3473).

### Mechanical disruption of organoids for ligand incubation

Organoids were cultured in organoid growth media with 10 µM Y-27632 for 3 days post-passage and cultured for an additional day in organoid growth media without Y-27632. Day 4 organoids were removed from Cultrex using 300µL of Cultrex Organoid Harvesting per dome and gently rotated at 4°C for 20 minutes. Organoids were then mechanically disrupted with a p1000 pipette tip fitted with a p10 pipette tip until crypts were dislodged from organoid structure. Crypts were spun down at 70 x g for 5 minutes at 4°C and resuspended in organoid growth media (- Y-27632) containing NOD1 or NOD2 ligands.

### CFU assay

Organoids were lysed in 0.1% sodium deoxycholate in D-PBS, and HeLa cells were lysed in 0.1% Triton X-100 in D-PBS following a 4-hour infection. Lysates were serially diluted in D-PBS, and each dilution was plated in duplicate onto TSB

agar plates supplemented with 1% Congo red for *S. flexneri*, and LB agar plates for *E. coli*. Plates were incubated overnight at 37°C, and colonies were counted the following day to determine bacterial load.

## Western blots

Following infection, organoids were harvested using Cultrex Organoid Harvesting Solution for 20 minutes and then lysed in RIPA buffer supplemented with protease inhibitor cocktail (150 mM sodium chloride, 50 mM Tris-HCl, 1% Igepal CA-630, 0.5% sodium deoxycholate, 0.1% SDS in ddH2O; Sigma, P8340). Lysates were centrifuged at 10,000 x g for 10 minutes at 4°C. Protein concentrations were normalized using the Pierce BCA protein assay kit according to the manufacture's protocol (ThermoFisher, 23225). The supernatants were mixed with Laemmli blue loading buffer and incubated at 95°C for 10 minutes. Samples were resolved on 10% acrylamide SDS-PAGE gels and transferred onto PVDF membranes. Membranes were blocked in 5% milk prepared in Tris-buffered saline with 0.1% Tween-20 (TBST) for 30 minutes at room temperature, followed by overnight incubation at 4°C with primary antibodies diluted in 5% milk in TBST. The next day, membranes were washed three times for 10 minutes each with TBST and incubated for 1 hour at room temperature with horseradish peroxidase (HRP)-conjugated secondary antibodies diluted in 5% milk in TBST. After three additional washes with TBST, HRP substrate was applied for five minutes at room temperature using Immobilon Classico Western HRP substrate for strong signals or SuperSignal West Dura Extended Duration Substrate for weaker signals (Sigma, WBLUC0500; ThermoFisher, 34076). Membranes were imaged using either a G:BOX Chemi XX6 imaging system and analyzed with GeneSys software or with the Invitrogen iBright 1500 Chemi luminescent reader and analysed with iBright imaging systems. The antibodies used include: Caspase-11 (Abcam, ab180673, 1:1000), GSDMD (Abcam, ab209845, 1:1000), α-Tubulin (Cell Signaling Technology, 2148S, 1:1000), Cleaved Caspase 3 (Cell Signaling Technology,9661S, 1:1000), Cleaved Caspase 8 (Cell Signaling Technology, 8592, 1:1000), GAPDH (Cell Signaling Technology, 5174S, 1:10,000) and Goat anti-Rabbit IgG (H+L) Secondary Antibody, HRP (ThermoFisher, 31460, 1:10,000).

## RT-qPCR

Total RNA was isolated using the GeneJET RNA purification kit following the manufacturer's instructions (ThermoFisher, K0732) and eluted in 30 µL of nuclease-free water. The isolated RNA was subsequently treated with the TURBO DNA-free Kit to remove contaminating genomic DNA according to the manufacturer's protocol (Invitrogen, AM190). Complementary DNA (cDNA) synthesis was performed using the High-Capacity cDNA Reverse Transcription Kit following the manufacturer's guidelines (ThermoFisher, 4374967). The resulting cDNA was diluted such that 5 ng was added to each quantitative PCR (qPCR) reaction, along with 0.5 µM of each forward and reverse primer in PowerUp SYBR Green Master Mix (ThermoFisher, A25778). qPCR was performed using the CFX384 Touch Real-Time PCR Detection System (Bio-Rad) with 40 amplification cycles at an annealing temperature of 60°C, followed by a melt curve analysis to verify product specificity. Target gene expression was normalized to housekeeping genes *Gapdh* or *Rpl19* ($2^{-\Delta CT}$). Primer sequences are listed in Table 1.

## Immunofluorescence

Following the passage of organoids, crypts were seeded onto coverslips placed in 24-well plates coated with Cultrex Pathclear Reduced Growth Factor Basement Membrane Extract and cultured as previously described. On day 4 of culture, organoids were infected and then fixed with 4% paraformaldehyde in D-PBS for 30 minutes. After fixation, organoids were washed three times with D-PBS and treated with 50 mM ammonium chloride for 30 minutes to quench autofluorescence. Organoids were permeabilized with 0.5% Triton X-100 in D-PBS for 30 minutes, followed by three additional washes with D-PBS. Blocking was performed for 30 minutes using 5% bovine serum albumin (BSA) in D-PBS, after which organoids were incubated overnight at 4°C with primary antibodies diluted in 1% BSA and 0.1% Triton X-100 in D-PBS. The following day, organoids were washed three times with D-PBS and incubated for one hour with secondary antibodies

**Table 1. qPCR primer sequences.**

| Gene | Forward sequence | Reverse sequence | Species |
|------|------------------|------------------|---------|
| *Gapdh* | 5'-GGAGCGAGACCCCACTAACA-3' | 5'-ACATACTCAGCACCGGCCTC-3' | Mouse |
| *Rpl19* | 5'-GCATCCTCATGGAGCACAT-3' | 5'-CTGGTCAGCCAGGAGCTT-3' | Mouse |
| *Lyz2* | 5'-ATGGAATGGCTGGCTACTATG-3' | 5'-GGTCTCCACGGTTGTAGTTT-3' | Mouse |
| *Pou2f3* | 5'-AATCTCACGTCTCCACCAAAG-3' | 5'-GGGAGTAAGGGCTGAAAGAAA-3' | Mouse |
| *Trmp5* | 5'-CCTGTAGACCACCTCAAATCAG-3' | 5'-CTAGGTCAGACAATCCTCCATATTC-3' | Mouse |
| *Tnfaip3* | 5'-GTAGAATCGGCTGCTTCCTATG-3' | 5'-CACCTCACTCCATCCCTATCT-3' | Mouse |
| *Cxcl1* | 5'-AGACCATGGCTGGGATTCAC-3' | 5'-AGTGTGGCTATGACTTCGGT-3' | Mouse |
| *Cxcl10* | 5'-TCAGGCTCGTCAGTTCTAAGT-3' | 5'-CCTTGGGAAGATGGTGGTTAAG-3' | Mouse |
| *Ccl20* | 5'-ACAGCCCAAGGAGGAAATG-3' | 5'-AGTCCACTGGGACACAAATC-3' | Mouse |
| *Nod1* | 5'-CCTTGCTGAGAGTCACCGTA-3' | 5'-CTGCCTTTCATTGCTGACC-3' | Mouse |
| *Nod2* | 5'-AAAGAGCTGCAGTTGAGGGAGGAA-3' | 5'-CACACATGGCCTTTGGTTTCCAGT-3' | Mouse |
| *Ripk2* | 5'-CGCTGCTCGACAGTGAAAGA-3' | 5'-AGTTTTAGACTGACACCAGTTACTT-3' | Mouse |
| *Isg15* | 5'-CTAGAGCTAGAGCCTGCAG-3' | 5'-AGTTAGTCACGGACACCAG-3' | Mouse |
| *Oasl1* | 5'-GTGCTCAAGGTACTCAAGGTAG-3' | 5'-CTGTGGAAACAGCTCAGGAA-3' | Mouse |
| *Nfkb2* | 5'-GCGGTGGAGACGAAGTTTAT-3' | 5'-AAGGCTTGCCATCCATTCT-3' | Mouse |
| *Pmepa1* | 5'-CTCCAGCGTAACTTTCCTTCTC-3' | 5'-GTCCCGCTAACGTGTGATAAT-3' | Mouse |
| *Plaur* | 5'-GCTTGAAGGATGAGGACTACAC-3' | 5'-AGTGAAAGGTCTGGTTGCTATG-3' | Mouse |
| *Rela* | 5'-GCTCAAGATCTGCCGAGTAAA-3' | 5'-GTCCCGTGAAATACACCTCAA-3' | Mouse |
| *Tnfaip8* | 5'-GTGCTTGGTGTGCCATTTC-3' | 5'-CAGTGGTTGGGTCTGTTACTT-3' | Mouse |
| *Casp4* | 5'-GCCACTTGCCAGGTCTACGAG-3' | 5'-AGGCCTGCACAATGACTT-3' | Mouse |
| *ChgA* | 5'- CCCGAAGTGACTTTGAGGAA-3' | 5'-ATGGCTGACAGGCTCTCTA-3' | Mouse |
| *Myt1* | 5'-GCTCTGATGATGACAAGGATG-3' | 5-CCTGTTCCAGAAGGCCTAAAT-3' | Mouse |
| *Rimbp2* | 5-GGCCTTTGAGCTTCCTTAGAT-3' | 5'-GTAAGCCCTTGGGTCAGTTAG-3' | Mouse |
| *Neurog3* | 5'-TACGACTTCCAGACGCAATTTA-3' | 5'-TGTCAAGCAGCAGTGGATAG-3' | Mouse |
| *Muc2* | 5'-ATGCCCACCTCCTCAAAGAC-3' | 5'-GTAGTTTCCGTTGGAACAGTGAA-3' | Mouse |
| *Alpi* | 5'-GGACATCGCCACTCAACTCA-3' | 5'-CACGTTTGCACCAGGTTCTG-3' | Mouse |
| *Lgr5* | 5'-TCGTGGTTCTGCATCTCCAT-3' | 5'-CGCTCCGGTATTGACCTGAT-3' | Mouse |
| *Neurod1* | 5'-GTCACTCCAAGACCCAGAAAC-3' | 5'-TGTACGAAGGAGACCAGATCA-3' | Mouse |
| *GAPDH* | 5'-GAGTCAACGGATTTGGTCGT-3' | 5'-GACAAGCTTCCCGTTCTCAG-3' | Human |
| *CXCL1* | 5'-CCTGCCCTTATAGGAACAGAAG-3' | 5'-AAGCGATGCTCAAACACATTAG-3' | Human |
| *IL-8* | 5'-TCTGCAGCTCTGTGTGAAGG-3' | 5'-ACTTCTCCACAACCCTCTGC-3' | Human |
| *TNFAIP3* | 5'-CGTCCAGGTTCCAGAACACCATTC-3' | 5'-TGCGCTGGCTCGATCTCAGTT-3' | Human |

at room temperature. After washing three times with D-PBS, organoids were stained with Hoechst dye and phalloidin for 15 minutes at room temperature (ThermoFisher, H1399; Sigma, P5282). Finally, organoids were washed again and mounted onto microscope slides using Dako Fluorescence Mounting Medium (Cedarlane, S302380-2), then left to dry in the dark for 1 hour at room temperature. Z-stack images (19–23 per sample) were captured and processed with orthogonal projections using ZEN imaging software. Slides were visualized on a ZEISS Laser-Scanning Confocal Microscope. Primary antibodies and stains are listed: pAb to *S. flexneri* was used at 1:100 (Abcam, ab65282), Ulex Europaeus Agglutinin I (UEA I) was used at 1:1000 (Vector Laboratories, RL-1062–2), Chromogranin A Monoclonal Antibody was used at 1:100 (ThermoFisher, MA5–13096), mouse-anti-ECadherin was used at 1:1000 (BD Bioscience, 610181), Anti-Serotonin antibody was used at 1:100 (Abcam, ab66047), anti-Rabbit IgG (H + L) Alexa Fluor 647 was use at 1:250 (ThermoFisher, A-31573), Rabbit anti-Goat IgG (H + L) Cross-Adsorbed Secondary Antibody, Alexa Fluor 488 was used at 1:250 (ThermoFisher, A11078), Goat anti-Rabbit IgG (H + L) Highly Cross-Adsorbed Secondary Antibody, Alexa Fluor 594 was used at

PLOS Pathogens

1:250 (ThermoFisher, A11037), and Goat anti-Mouse IgG (H + L) Highly Cross-Adsorbed Secondary Antibody, Alexa Fluor 488 was used at 1:250 (ThermoFisher, A11029).

## ELISA

Quantitative measurements of CXCL1 were performed on enteroid supernatants using an ELISA kit according to the manufacturer's instructions (R&D Systems, DY453).

## Cellular ATP

Cellular ATP was measured after infection with CellTiter-Glo Luminescent Cell Viability Assay according to the manufacturer's instructions (Promega, G7571).

## Sequencing and analysis

RNA isolation was performed as previously described. Sequencing libraries were prepared and sequenced on an Illumina NovaSeq platform using an S4 flow cell, generating 150 base pair paired-end reads with a depth of 20–30 million reads per sample. Raw sequencing reads were aligned to the reference genome using STAR aligner [58]. Differential gene expression analysis was conducted using DESeq2 [59]. Functional enrichment analysis was performed with ClusterProfiler [60]. Heatmaps illustrating expression patterns were generated using the pheatmap package in R.

Single cell RNAseq analysis was performed as previously described [35]. Bulk RNA sequencing data can be accessed from the GEO repository under the accession number GSE305881.

## Supporting information

**S1 Fig. GAPs do not play a major role in bacterial uptake in *Nlrc4*⁻/⁻ IECs.** (**A**) Confocal immunofluorescence of *Nlrc4*⁻/⁻ enteroids infected with M90T or BS176. Arrows indicate bacteria colocalizing with UEA1⁺ cells. Scale bar represents 20 um. (**B**) Confocal immunofluorescence of *Nlrc4*⁻/⁻ enteroids incubated with fluorescent beads for 4 hours. Arrows indicate beads colocalizing with UEA1⁺ cells. Scale bar represents 20 um. (**C**) CFU analysis of *Nlrc4*⁻/⁻ enteroids pretreated with either ISX-9 (40uM) or DAPT (10uM) for 48 hours, then infected with M90T or BS176. (**D**) qPCR validation of cell type-specific marker in DAPT- and ISX-9- treated. *Nlrc4*⁻/⁻ enteroids. Statistical analysis in panel **C** was performed using Two-way ANOVA, and statistical analysis in panel **D** was performed using One-way ANOVA. Data represent mean ± SD, with $p < 0.05$ considered statistically significant. All infections were performed at an MOI of 50 for 4 hours. Data are representative of at least three independent biological replicates.
(TIF)

**S2 Fig. Transcriptomic profiling of *Nlrc4*-/-*Ripk2*+/+ vs *Nlrc4*-/-*Ripk2*-/- murine intestinal organoids following *S. flexneri* M90T infection.** (**A**, **B**) Volcano plots of differentially expressed genes plotted by FDR correction between *Nlrc4*⁻/⁻*Ripk2*⁺/⁺ and *Nlrc4*⁻/⁻*Ripk2*⁻/⁻ ileal organoids treated with M90T (**A**) or M90T-filtered supernatant (**B**). (**C**) Volcano plot illustrating gene expression between *Nlrc4*⁻/⁻*Ripk2*⁺/⁺ vs *Nlrc4*⁻/⁻*Ripk2*⁻/⁻ murine ileal organoids infected with M90T using unadjusted *p*-values; no genes reached statistical significance after correction for multiple testing using FDR adjustment. Genes selected for subsequent validation by quantitative PCR are indicated in red. Data are representative of four independent biological replicates.
(TIF)

**S3 Fig. Apical stimulation with NOD1 and NOD2 ligands does not elicit a pro-inflammatory response in murine intestinal organoids.** (**A**) qPCR analysis of WT murine ileal organoids were mechanically disrupted to expose the apical surface and incubated for 2 hours with NOD1 (C12-iE-DAP, Tri-DAP) or NOD2 (L18-MDP) ligands – each at 1 µg/mL.

(**B**) qPCR analysis of apical-out organoids were similarly stimulated with C12-iE-DAP or L18-MDP (1 µg/mL) for 4 hours TNFα (10 ng/mL, 4 hours) was included as a positive control. Statistical analysis for panels **A** and **B** was performed using One-way ANOVA. Data represent mean±SD, with $p < 0.05$ considered statistically significant. Data are representative of at least three independent biological replicates.
(TIF)

**S4 Fig. Diverse stimuli reveal EEC-associated response in *Nlrc4*[-/-] enteroids.** (**A**) GSEA was performed using transcriptomic data from *Nlrc4*[-/-] enteroids infected with *S. flexneri* M90T at an MOI of 50 compared to UI controls. Subtype-specific EEC gene signatures were defined according to Gehart et al. [39]. (**B**) qPCR analysis of *Nlrc4*[-/-] enteroids infected with DH5α *E. coli* transformed with an afimbrial adhesin protein AfaE at an MOI of 50 for 4 hours. (**C**) Volcano plot of differentially expressed genes between TSB control and M90T supernatant-treated *Nlrc4*[-/-] Intestinal organoids. Statistical analysis in panel **B** was performed using one-way ANOVA. Data represent mean±SD, with $p < 0.05$ considered statistically significant. Data are representative of at least three independent biological replicates.
(TIF)

**S5 Fig. Proposed model of the intestinal epithelial responses to *Shigella flexneri*.** In WT ileal organoids, recognition of the *S. flexneri* T3SS triggers inflammasome activation and epithelial pyroptosis. In *Nlrc4*-deficient organoids, pyroptosis does not occur in response to M90T alone; instead, invasive and non-invasive bacteria are sampled within a tolerogenic program associated with reduced inflammation, alongside increased differentiation toward enteroendocrine lineages. Figure was created in BioRender. PHILPOTT, D. (2026) https://BioRender.com/yz630ln.
(TIF)

## Acknowledgments

We thank Dr. Richard Flavell for the *Ripk2*[-/-] mice, Dr. Russel Vance for the *Nlrc4*[-/-] mice, Dr. Catherine O'Brien for the CSC28 cells, and Dr. Jeremy Mogridge for the NeedleTox and PA used in this study.

## Author contributions

**Conceptualization:** Marry Nissan, Dana J Philpott, Stephen E. Girardin.

**Funding acquisition:** Dana J Philpott, Stephen E. Girardin.

**Investigation:** Marry Nissan, Adrienne Ranger, Justin J Meade, Victoria Gillmore, Maija E Lehn, Liliane Cabral-Fernandes, Derek KL Tsang.

**Methodology:** Marry Nissan, Adrienne Ranger, Justin J Meade, Victoria Gillmore, Maija E Lehn, Liliane Cabral-Fernandes, Derek KL Tsang.

**Resources:** Liliane Cabral-Fernandes, Scott D Gray-Owen, Dana J Philpott, Stephen E. Girardin.

**Supervision:** Scott D Gray-Owen, Dana J Philpott, Stephen E. Girardin.

**Writing – original draft:** Marry Nissan, Stephen E. Girardin.

**Writing – review & editing:** Marry Nissan, Adrienne Ranger, Justin J Meade, Maija E Lehn, Liliane Cabral-Fernandes, Derek KL Tsang, Dana J Philpott, Stephen E. Girardin.

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
