## [Decision Letter · Decision Letter 0]

30 Sep 2025

Host response to bacteria induces a shift towards the enteroendocrine cell lineage in the murine enteroid model

PLOS Pathogens

Dear Dr. Girardin,

Please submit your revised manuscript within 60 days Nov 29 2025 11:59PM. If you will need more time than this to complete your revisions, please reply to this message or contact the journal office at plospathogens@plos.org. Please include the following items when submitting your revised manuscript:

We look forward to receiving your revised manuscript.

Kind regards,

Nicholas J Mantis

Academic Editor

PLOS Pathogens

Matthew Wolfgang

Section Editor

PLOS Pathogens

Editor-in-Chief

PLOS Pathogens

Michael Malim

Editor-in-Chief

PLOS Pathogens

orcid.org/0000-0002-7699-2064

**Journal Requirements:**

If you are using the free assets from BioRender, we are unable to publish these images as they are licenced under a stricter licence than CC BY 4.0. In this case we ask you to remove the BioRender images and replace them with open source alternatives.

See these open source resources you may use to replace images / clip-art:

- https://bioart.niaid.nih.gov/

- https://bioicons.com/

- https://healthicons.org/

- https://scidraw.io/

- https://reactome.org/icon-lib

- https://www.phylopic.org/images

  5) "Thank you for stating "Bulk RNA sequencing data can be accessed from the GEO repository under the accession number GSE305881." Please note that, though access restrictions are acceptable now, your entire minimal dataset will need to be made freely accessible if your manuscript is accepted for publication. This policy applies to all data except where public deposition would breach compliance with the protocol approved by your research ethics board.  6) Please amend your detailed Financial Disclosure statement. This is published with the article. It must therefore be completed in full sentences and contain the exact wording you wish to be published.1) State what role the funders took in the study. If the funders had no role in your study, please state: "The funders had no role in study design, data collection and analysis, decision to publish, or preparation of the manuscript.". Note: If the reviewer comments include a recommendation to cite specific previously published works, please review and evaluate these publications to determine whether they are relevant and should be cited. There is no requirement to cite these works unless the editor has indicated otherwise.

**Reviewers' Comments:**

Reviewer's Responses to Questions

**Part I - Summary**

Reviewer #1: Girardin and colleagues present an interesting study that investigates inflammatory signaling pathways and the impact of S. flexneri on mouse intestinal organoid-derived cells. A strength of this work is the use of a more physiologically relevant model such as mouse intestinal organoids and comparing the results obtained to human epithelial cell lines. However, the authors should take care when making broad statements, as the findings in mouse cells may not be fully recapitulated in human cells. Including a mouse cell line in some of the experiments may be a useful comparison.

The authors have presented some very interesting and novel findings here. The data showing that internalisation and response to an avirulent strain of Shigella is unexpected. As too is the response to bacteria-free supernatants. Another strength of this work is the RNA sequence analysis, which revealed a shift towards more differentiated EEC cells. However, these findings have not been fully validated in qPCR results and require follow up experiments.

A point to consider is that these experiments have been performed using mouse ileal organoids while the Shigella burden is in the colon. This may impact the findings made, as Shigella may not replicate as efficiently in these cells. The authors should address this as a potential limitation of the work, although the data presented here are very valuable nonetheless, and I believe of interest to a broad audience.

Reviewer #2: In this manuscript, the authors used murine intestinal organoids to study the host response to Shigella flexneri infection. They first confirmed that a Nlrc4-dependent Gsdmd cleavage occurred upon Shigella infection of wt organoids using a virulent strain and suggested that murine organoids uptake bacterial ligands from the virulent strain directly from the supernatant, independently of invasion (Fig. 1). Then, they tried to study the role of Nod1/2 in this infection organoid model (Fig. 2), but concluded that murine organoids are not responsive to Nod1/2 ligands (Fig. 3), making the results of infected Nod1/2 organoid a bit irrelevant. Next, they observed an infection-dependent, but virulence-independent transcriptional change upon infection/treatment of organoids with different Shigella strains/supernatant (Fig. 4). Finally, they suggested that infection led to an enrichment of late enteroendocrine cells in a virulence-independent fashion using RNA-Seq analysis (Fig. 5). The manuscript is well written and many of the data presented in this manuscript are interesting. However, it is quite difficult to connect the dots and most of the points raised by the authors are not supported by enough evidence and remain unanswered at the end of the manuscript. I believe that the manuscript would gain in clarity without the Nod1/2 part and with more data on the uptake/invasion mechanisms and EEC maturation (maybe using a time kinetic). In addition, the addition of a summary figure at the end of the manuscript would help the reader.

**Part II – Major Issues: Key Experiments Required for Acceptance**

Please use this section to detail the key new experiments or modifications of existing experiments that should be absolutely required to validate study conclusions.required to validate study conclusions.

Reviewer #1: - On line 133 the authors state that there are differences in bacterial uptake and replication, however only one timepoint has been shown. A timecourse experiment is required to show replication of S. flexneri in organoids. It is important for interpretation of the model to understand whether the virulent M90T is able to set up a productive infection in these cells. It may be that the CFU represent bacterial uptake rather than replication, and this is why M90T and BS176 have similar values. Could it also be possible that bacteria that access the organoid lumen are protected from gentamicin? Testing the uptake of E. coli or labelled beads would also be really useful to understand whether the phenomenon of particle uptake is specific to Shigella.

- What is the lower band in Figure 1H? It could be p20 subunit of GSDMD indicating that there has been cleavage and activation. Other assays for cell death or inflammasome activation are required to understand whether other pathways are activated. For example, caspase 1 cleavage may be occurring here. Furthermore, it is too broad to state on line 145 that ‘other inflammasome pathways…remain inactive in this context’.

- The qPCR validation of RNA sequencing presented here is not very robust and does not fully support the statements made by the authors. Other methods to assess inflammatory signalling such as cytokine analysis (Fig 4) or detection of differentiation markers by other methods such as by microscopy or flow cytometry (Fig 5) would further support the claims made by the authors. It would also be useful to know whether there is any cell death in their model, to understand whether the shift to terminally differentiated EECs is a consequence of the bacteria on differentiation or of loss of less differentiated cells

Reviewer #2: The title is misleading as it is not supported by enough evidence in this paper. Only the last figure is related to this point and the data presented derive mostly from one RNA-Seq analysis. They haven’t been confirmed independently. I’d suggest adjusting the title or the content of the manuscript.

L.136-140: if Caspase 11 is activated and therefore cleaved, shouldn’t the level of full-length caspase 11 decrease? I suggest showing the picture of the full membrane for all caspase 11 western blots to allow the visualisation of cleaved caspase-11 in addition to full-length caspase-11. And in Fig. 1H, the disappearance of uncleaved Gsdmd correlates with the appearance of a 20KDa band in interferon-treated infected organoids. How do the authors explain that? Another Gsdmd antibody could be used to confirm the findings. Additionally, a faint band is seen at for the N-term cleaved Gsdmd at around 35KDa for the Ifng-treated M90T infected organoids.

L.142: Shigella with a cytosolic reporter plasmid could be used to see if the bacteria reach the cytosol upon infection. In addition, as the authors suggested cytosolic uptake of PAMPs by the enteroids (L.301, L.305-306), one could assume that LPS uptake could as well occur. A western blot analysis of LPS stimulated organoids, as it has been done with NeedleTox, would be nice.

Fig. 5: please confirm your findings using immunofluorescence (+ quantification) with known EEC and, if available, late EEC markers.

L.307-314: the authors suggested a role for GAPs and EECs in the uptake of PAMPs and bacteria. Using Goblet (e.g. Muc2) and EEC (e.g. Chromogranin A) markers and a time kinetic, it should be possible to observe early bacterial invasion/PAMPs uptake in this organoid model by immunofluorescence. Such mechanistic datasets would strengthen the manuscript.

**Part III – Minor Issues: Editorial and Data Presentation Modifications**

Reviewer #1: - There is no indication of number of biological replicates that have been performed, which may cast some doubt over how reproducible the findings are. This information would be useful throughout the manuscript

- Error on line 123 ‘ligand cytosolic uptake of the ligand’

- For Fig 2A it is unclear what is meant by ‘relative mRNA expression’ when referring to RNA sequencing results, could the authors please clarify this

- Given differences between inflammasome activation and susceptibility to Shigella between humans and mice, the authors should be careful when making broad statements. Eg line 143, ‘mouse’ should be added before ‘intestinal epithelium’ for clarity.

- While the rationale for using RIPK2 KO organoids to study NOD signalling is sound, it is an overstatement to say that ‘NOD1 and NOD2 do not alter the transcriptome…’ The authors should modify the language used to reflect that the investigation was of RIPK2-dependent effects.

- Fig 4A,B and 5C the text is too small

- Statement on line 231 ‘This finding challenges the prevailing model…’ is an overstatement, as it is well established that inflammatory signalling can be inhibited by pathogens during enteric infection

Reviewer #2: Figure 2-3: the authors generated Nlrc4-/-Ripk2-/- mice to generate organoids. However, they concluded that wt organoids do not respond to Nod1/2 ligands in Fig. 3, making the results presented in Fig. 2 a bit irrelevant. These 2 datasets have probably been generated in this order, but as such do not bring much to the main story. I suggest removing them or presenting them at best as supplementary data to keep the flow of the manuscript.

Maybe out of scope for this paper as the authors did not use any mouse infection in this manuscript, but are Nlrc4-/-Ripk2-/- mice more or less susceptible to infection than the single ko mice?

Fig. 4B: the labelling “increase with” and “decrease with” should be adapted to the conditions. The figure legend is unclear. It sounds as if untreated organoids were compared to supernatant-treated organoids, but it doesn’t fit with what’s mentioned in the text. Please clarify. In addition, it would be nice to show the volcano plots of DE genes between supernatant-treated and untreated organoids to corroborate the conclusion that the effect is specific to infection.

Fig. 4C: Organoid infection led as well to the upregulation of many genes. Please analyse those genes as well and show the terms associated with these infection-induced genes. The data currently presented are not enough to state that “this finding challenges the prevailing model of inflammatory IEC signalling during enteric infection”. The authors would need to show a GSEA analysis including all up and down regulated genes to make such claim.

Fig. 5B: is this analysis the counterpart of Fig. 4C? If yes, please present the data using the same layout, number of top pathway analysed and criteria (organised by gene ratio in Fig. 4C).

Fig. 4E: the authors handpicked only 6 genes to reach their conclusion. It would be better to analyse the difference between the virulent and non-virulent strain using RNA-Seq.

L.123, remove the extra “ligand”

PLOS authors have the option to publish the peer review history of their article (what does this mean? ). If published, this will include your full peer review and any attached files.). If published, this will include your full peer review and any attached files.

**Do you want your identity to be public for this peer review?** For information about this choice, including consent withdrawal, please see our For information about this choice, including consent withdrawal, please see our Privacy Policy ..

Reviewer #1: No

Reviewer #2: No

**Figure resubmission:**

**Reproducibility:**



---

## [Decision Letter · Decision Letter 1]

2 Mar 2026

PPATHOGENS-D-25-02070R1

Host response to bacteria induces a shift towards the enteroendocrine cell lineage in the murine enteroid model

PLOS Pathogens

Dear Dr. Girardin,

Thank you for submitting your manuscript to PLOS Pathogens. Reviewer 1 requested your attention to several minor issues that they wished to have clarified before final acceptance. Please see comments below.

We look forward to receiving your revised manuscript.

Kind regards,

Nicholas J Mantis

Academic Editor

PLOS Pathogens

Matthew Wolfgang

Section Editor

PLOS Pathogens

Sumita Bhaduri-McIntosh

Editor-in-Chief

PLOS Pathogens

orcid.org/0000-0003-2946-9497

Michael Malim

Editor-in-Chief

PLOS Pathogens

orcid.org/0000-0002-7699-2064

**Additional Editor Comments :**

Dear Dr. Girardin, Reviewer 1 requested your attention to several minor issues that they wished to have clarified before final acceptance. Please see comments below. Thank you, Nicholas Mantis

**Journal Requirements**

1) Thank you for stating that "Bulk RNA sequencing data can be accessed from the GEO repository under the accession number GSE305881."  We noted that it is currently private and is scheduled to be released on Aug 01, 2026 .  Please note that though access restrictions are acceptable now, your entire minimal dataset will need to be made freely accessible if your manuscript is accepted for publication. This policy applies to all data except where public deposition would breach compliance with the protocol approved by your research ethics board.

**Reviewers' Comments:**

Reviewer's Responses to Questions

**Part I - Summary**

Reviewer #1: I commend the authors on their responses to reviewer comments and the additional experiments performed which add value to the manuscript and clarify some of the observations made. I am overall satisfied with responses to my comments but I do still think there are some minor issues with the manuscript.

Reviewer #2: The authors have addressed many of my comments and I’m overall satisfied with the current version of the manuscript. I only have minor points that I picked up while reviewing the manuscript.

L.777: add abbreviations used for the UMAP in the figure legend.

L.787: I believe the authors meant to say that Panel C was created with Biorender.

L.802/803: the panels are E and F, not A and B.

L.807: the relative expression of Nod1 is not shown.

**Part II – Major Issues: Key Experiments Required for Acceptance**

Please use this section to detail the key new experiments or modifications of existing experiments that should be absolutely required to validate study conclusions.required to validate study conclusions.

Reviewer #1: N/A

Reviewer #2: (No Response)

**Part III – Minor Issues: Editorial and Data Presentation Modifications**

Reviewer #1: 1. The authors stated in their responses that figure legends have been updated to include biological replicate information but I still can't find this information.

2. Caspase cleavage at baseline levels seems a bit high (Fig 1g), it would be good if the authors could comment on this

3. For the comment on figure 2G (lines 177-179), the authors state that there was no GSDMD cleavage, but it seems there is with IFNg priming?

4. Line 277, should this be reference to Fig 4H, I instead?

5. some typographical errors in discussion, line 376 'findings' should be 'finding', line 405 'we explores'

6. Sup fig 5 is not referred to in the text

Reviewer #2: (No Response)

PLOS authors have the option to publish the peer review history of their article (what does this mean? ). If published, this will include your full peer review and any attached files.). If published, this will include your full peer review and any attached files.

**Do you want your identity to be public for this peer review?** For information about this choice, including consent withdrawal, please see our For information about this choice, including consent withdrawal, please see our Privacy Policy ..

Reviewer #1: No

Reviewer #2: No

**Figure resubmission:**
---

## [Editor Report · Decision Letter 2]

5 Mar 2026

Dear Dr. Girardin,

We are pleased to inform you that your manuscript 'Host response to bacteria induces a shift towards the enteroendocrine cell lineage in the murine enteroid model' has been provisionally accepted for publication in PLOS Pathogens.

Best regards,

Nicholas J. Mantis

Academic Editor

PLOS Pathogens

Matthew Wolfgang

Section Editor

PLOS Pathogens

Sumita Bhaduri-McIntosh

Editor-in-Chief

PLOS Pathogens

orcid.org/0000-0003-2946-9497

Michael Malim

Editor-in-Chief

PLOS Pathogens

orcid.org/0000-0002-7699-2064
---

## [Editor Report · Acceptance letter]

Dear Dr. Girardin,

We are delighted to inform you that your manuscript, "Host response to bacteria induces a shift towards the enteroendocrine cell lineage in the murine enteroid model," has been formally accepted for publication in PLOS Pathogens.

Best regards,

Sumita Bhaduri-McIntosh

Editor-in-Chief

PLOS Pathogens

orcid.org/0000-0003-2946-9497

Michael Malim

Editor-in-Chief

PLOS Pathogens

orcid.org/0000-0002-7699-2064